# Numerical Simulation of Hydraulic Fractures Breaking through Barriers in Shale Gas Reservoir in Well YS108-H3 in the Zhaotong Shale Gas Demonstration Area

**Shasha Sun [1],\*, Xinyu Yang [2], Yun Rui [3], Zhensheng Shi [1], Feng Cheng [1], Shangbin Chen [2], Tianqi Zhou [1], Yan Chang [1] and Jian Sun [1]**

1   PetroChina Research Institute of Petroleum Exploration and Development, Beijing 100083, China; shizs69@petrochina.com.cn (Z.S.); chengfeng3749@petrochina.com.cn (F.C.); zhoutianqi@petrochina.com.cn (T.Z.); changyan05@petrochina.com.cn (Y.C.); sunjiansdkd@163.com (J.S.)
2   School of Resources and Geosciences, China University of Mining and Technology, Xuzhou 221000, China; yangxinyu_99@163.com (X.Y.); shangbinchen@163.com (S.C.)
3   PetroChina Zhejiang Oil and Gas Field Company, Hangzhou 310000, China; ruiy85@petrochina.com.cn
\*   Correspondence: sunshasha501@163.com

**Abstract:** Estimating the effectiveness of hydraulic fracturing in the context of the incrfease in the shale gas demand is of great significance for enhancing shale gas production, which aims to substantially reduce fossil energy consumption and $CO_2$ emissions. The Zhaotong national shale gas demonstration zone has complex stress structures and well-developed fracture zones, and thus it is challenging to achieve targeted reservoir segment transformation. In this paper, we construct and optimize the geometry of hydraulic fractures at different pressures considering the upper and lower barriers in hydraulic fracturing simulation experiments and numerical modeling. The numerical simulation results show that the pore pressure exhibits a stepped pattern around the fracture and an elliptical pattern near the fracture tip. During the first time of injection, the pore pressure rapidly increases to 76 MPa, dropping sharply afterward, indicating that the fracture initiation pressure is 76 MPa. During the fracture propagation, the fracture length is much greater than the fracture height and width. The fracture width is larger in the middle than on the two sides, whereas the fracture height gradually decreases at the fracture tip in the longitudinal direction until it closes and is smaller near the wellbore than at the far end. The results revealed that the fracture width at the injection point reached the maximum value of 9.05 mm, and then it gradually decreased until the fracture width at the injection point dropped to 6.33 mm at the final simulation time. The fracture broke through the upper and lower barriers due to the dominance of the effect of the interlayer principal stress difference on the fracture propagation shape, causing the hydraulic fracture to break through the upper and lower barriers. The results of the physical simulation experiment revealed that after hydraulic fracturing, multiple primary fractures were generated on the side surface of the specimen. The primary fractures extended, inducing the generation of secondary fractures. After hydraulic fracturing, the width of the primary fractures on the surface of the specimen was 0.382–0.802 mm, with maximum fracture widths of 0.802 mm and 0.239 mm, representing a decrease of 70.19% in the maximum fracture width. This work yielded an important finding, i.e., the urgent need for hydraulic fracturing adaptation promotes the three-dimensional development of a gas shale play.

**Keywords:** hydraulic fracturing; fracture breakthrough; numerical simulation; Zhaotong Shale Gas Demonstration Area



## 1. Introduction

Shale gas is an important unconventional natural gas resource, following coalbed methane and tight sandstone gas [1]. It serves as a significant supplement to conventional oil and gas resources. Shale gas has the characteristics of a long extraction life, extended

production cycles, shorter hydrocarbon migration distances, and large gas-bearing areas [2]. In the 20th century, shale gas in the United States underwent a process from discovery to industrial production and then to high-speed production through technological transformation [3]. China has an enormous potential for shale gas resources. Effective exploration and development of shale gas resources can accelerate the construction of a clean, low-carbon, safe, and efficient modern energy system, improve air quality, and achieve green and low-carbon development [4]. In 2017, the China Petroleum Company conducted gas testing in old wells Y1 and Y102 in the Zhaotong shale gas demonstration zone in the Zhejiang Oilfield. At present, a demonstration area with a production capacity of $20 \times 10^8$ m$^3$ has been built in the middle-deep layer of the Huangjin dam and the shallow solar anticline area in the Zijin dam [5].

The core techniques for shale gas development are horizontal drilling and hydraulic fracturing, and volume fracturing is one of the key technical aspects of hydraulic fracturing [6]. Volume fracturing refers to the process of continuously expanding natural fractures and inducing shear slip in brittle rocks during hydraulic fracturing, resulting in an interlocking network of natural and artificial fractures [7]. This increases the stimulated volume and enhances the initial production and ultimate recovery rates. To achieve effective volume fracturing, it is necessary for the shale to have well-developed fractures or bedding, and the natural fractures or bedding needs to be oriented in alignment with the minimum principal stress direction [8]. This ensures that the hydraulic fractures are perpendicular to the natural fractures, facilitating the formation of an interconnected network of fractures. Additionally, the brittleness of the shale plays a crucial role in the type of fracturing damage induced [9]. Shales with higher brittleness coefficients are more prone to shear failure during fracturing, rather than forming a single fracture. This favors the development of complex interconnected fracture networks, thereby increasing the distribution range and volume of fractures. There are significant differences in the fracturing mechanisms of horizontal wells, vertical wells, and conventional directional wells [10]. Horizontal wells have their own complexity and uniqueness. The drilling environment encountered in horizontal wells is more complex, and the initiation pressure required for hydraulic fracturing is much higher than the fracturing pressure in vertical wells [11]. It is common for fractures to fail to propagate, leading to fracturing failure. In-depth research on the fracturing initiation mechanisms in horizontal wells and identification of rational initiation patterns are essential prerequisites for successful hydraulic fracturing operations in horizontal wells [11].

Recent shale gas reforming practices in China have further confirmed that there are cases of limited or excessive fracture height extension during the construction process, making it difficult to achieve efficient reforming process optimization and design. Fracture monitoring results have revealed that it is difficult to control the fracture height in shale reservoirs, and the fractures between different layers easily connect with each other, which increases the filtration loss of the fracturing fluid in the longitudinal direction and is not conducive to fracture extension, thus limiting the scope of fracturing modification. Hydraulic fracturing fractures usually expand along the direction of the maximum principal stress. There are a large number of natural fractures and laminae in shale, and the structure is clearly anisotropic, resulting in the interaction behaviors of hydraulic fracturing fractures such as penetration, capture, steering, and offset along the matrix, natural fractures, and laminae. It is difficult to accurately predict the expansion pattern of the fracturing fracture network and the spatial distribution pattern. In the actual hydraulic fracturing construction process, the fractures extend not only horizontally but also in the longitudinal height direction, and the fracture length and fracture height geometries increase. The competitive fracture initiation and extension mechanism among multiple fractures in horizontal shale wells, the vertical extension of fractures under stratification conditions, the balanced initiation and extension of multiple fractures, and the revelation of hydraulic fracture extension patterns in unconventional reservoirs are the key issues for achieving efficient development

of shale reservoirs; and, therefore, there is an urgent need to carry out research on the extension law of shale hydraulic fracture through the layers.

Hydraulic fracturing technology can create fractures to increase permeability and accelerate the movement of gas and water in shale reservoirs [12]. It should be noted that the geometry of fracture extension during the fracturing construction process is one of the main factors affecting the fracturing effect [13]. To deeply analyze the influencing factors of the hydraulic fracturing process and determine the rules of hydraulic fracturing fracture propagation, scholars have conducted a large number of numerical simulations and experimental studies [14].

Previous studies have demonstrated that the stress distribution around the wellbore in horizontal wells plays a decisive role in fracture initiation [15]. Additionally, the in situ stress, completion methods, phase angle and diameter of the jet nozzles, and depth of perforations all have certain influences on fracture initiation and propagation [16]. During hydraulic fracturing in horizontal wells, as multiple fractures extend, the fracture size, shape, and orientation vary. Interactions between fractures occur, and appropriate inter-cluster spacing, rock parameters, and in situ stress are obtained. Increasing the degree of fracture damage will maximize the effective area and achieve maximum production.

In the actual hydraulic fracturing process, fractures not only propagate and expand horizontally but also extend vertically. Both the length and height of the fractures increase in the geometric dimensions [17]. Accurate prediction of the geometric dimensions and propagation range of the fractures is of great significance for determining reasonable fracturing construction parameters and predicting oil and gas production [18]. Understanding the conditions and criteria for fracture propagation is the foundation for analyzing the fracture expansion patterns in hydraulic fracturing in horizontal wells [19]. Typically, numerical simulation models are established to guide the study of fracture propagation patterns by simulating the extension of fractures [20].

Currently, numerical simulation methods for hydraulic fracturing mainly include displacement discontinuity methods (DDMs) based on continuum mechanics, the finite element method (FEM), the extended finite element method (XFEM), and discrete element method (DEM) based on discontinuum mechanics [21]. Currently, the FEM and XFEM have been widely used in many hydraulic fracturing engineering designs. The FEM, as a traditional and classical numerical computation method, has significant advantages in solving nonlinear mechanics problems and complex stress–strain problems [22]. It uses mesh reconstruction methods to simulate fracture propagation by aligning the fracture boundaries with the mesh element boundaries. It has been widely applied in numerical simulations of hydraulic fracturing. Hunsweck used the FEM to establish a two-dimensional fluid–solid coupling hydraulic fracturing model that considers rock deformation and the fluid flow inside the fracture [23]. This model simulates the delay phenomenon of fluid at the fracture tip during the fracture propagation process [23]. Bao et al. introduced a reduced algorithm into the FEM to simulate hydraulic fracturing problems, significantly reducing the computational cost by only considering the degrees of freedom of the nodes on the upper and lower surfaces of the fracture for fluid–solid coupling iterations [24]. Omidi et al. used an adaptive mesh to establish a spatially and temporally discontinuous Galerkin FEM to simulate the initiation and propagation of hydraulic fractures along arbitrary paths [25]. Kim et al. used the FEM to study tension hydraulic fractures that propagate and extend vertically in shale gas reservoirs and found that the occurrence of tension fractures exhibits temporal discontinuity [26]. Pan Linhua et al. addressed the problem of volume fracture propagation in shale reservoirs [27]. They established a three-dimensional FEM for hydraulic fracturing of volume fractures in shale reservoirs based on the basic coupled fluid–solid equations and the principles of the damage mechanics [27]. Ma et al. simulated the morphology and propagation mechanism of fractures and has achieved good agreement with laboratory test results [28]. This study focused on the dynamic responses of rock and PMMA (polymethyl methacrylate) induced by tamped spherical detonation [28]. By conducting mini-chemical explosion tests and utilizing a four-dimensional lattice spring model, they investigated

the behavior of rock under dynamic loading conditions [28]. Huang et al. explored the phenomenon of soil-water inrush-induced damage to shield tunnel linings and presented a comprehensive analysis of stabilization measures [29]. It would provide a practical example of the challenges faced in tunnel engineering and the corresponding mitigation strategies employed [29]. It would also demonstrate the importance of considering geotechnical factors in deep engineering projects [29].

Previous studies have mainly focused on the propagation of fractures within a reservoir, but there is a lack of research on the influences of the upper and lower barriers on fracture propagation in shale. In this paper, the shale gas reservoir in the Zhaotong Shale Gas Demonstration Area is investigated using the soil module of the ABAQUS 2022 software. A three-dimensional hydraulic fracture propagation model is established based on the three-dimensional cohesive element method and field measurement data. The changes in the pore pressure, overall morphology of fractures, and reasons for breaking through the upper and lower barriers during the fracture propagation process in a shale reservoir are analyzed. Using the ABAQUS finite element software, a model is established to investigate the interaction between hydraulic fractures and natural fractures. The propagation behavior and expansion patterns of hydraulic fractures intersecting with natural fractures under different stress differentials are determined. This research provides a more realistic understanding of fracture propagation behavior and theoretical guidance for hydraulic fracturing in reservoirs with developed natural fractures. Hydraulic fracturing fluids for shale gas production are composed of a sand-based proppant, sufficient water, and a small amount of chemical additives, of which the first two and chemical additives make up 99.5% and 0.5% of the overall composition, respectively. The pressure is transmitted through the fracture when it is in the fracturing state. With the continuous progress of exploration and development technology, the adverse impact of shale gas hydraulic fracturing on the ecological environment has been greatly improved, but it is still unavoidable. In this paper, through numerical simulations of shale gas hydraulic fracturing, we further determine the amount of water used in the fracturing process and the amount of fracturing fluid returned, and then we assess the avoidance of the irreversible adverse impacts on the ecological environment caused by shale gas hydraulic fracturing from the source, promoting the green and sustainable development of shale gas. The main contribution of this work lies in its direct relevance to sustainability. By enhancing our understanding of hydraulic fractures breaking through barriers, we contribute to the efficient extraction of shale gas resources, which can serve as a transitional energy source towards a greener future. The numerical simulation of shale gas well fracturing scheme in this paper can provide a more solid scientific basis for the optimization of a shale gas fracturing scheme at a later stage, which, in turn, can reduce the consumption of freshwater resources, lower the pressure of water supply, avoid surface and groundwater pollution, and mitigate atmospheric pollution.

## 2. Geologic Setting

The Zhaotong Shale Gas Demonstration Area is located at the intersection of Yunnan, Guizhou, and Sichuan provinces [30]. It is situated in the Wumeng Mountains transitional zone from the Yungui Plateau to the Sichuan Basin, and its geological structure is controlled by the overlap between the western edge of the Sanjiang orogenic belt and the southeastern front of the Jiangnan–Xuefeng structural belt in the Yangtze Block [30]. The main part of the area is located in the central-western part of the Weixin depression in the northern Sichuan–Guizhou Basin [31]. The current structural morphology of the study area is influenced by multiple tectonic movements and has complex surface structures with typical mountainous features [31]. Several northeast-trending synclines and anticlines are developed in the area, arranged in alternating strips, and the preservation of shale gas is clearly controlled by the distribution of the structures and faults [32].

The YS108-H3 horizontal well group studied in this paper is located in the southern margin of the low and steep fold belt in southern Sichuan in the Sichuan platform downwarp, adjacent to the Dianqianbei Depression (Figure 1). This horizontal well group is

located in the southern flank of the Jianwu anticline. The strata of the axis part are Jurassic strata, and the wing is composed of the Triassic Leikoupo Formation, Jialingjiang Formation, Tongjiazi Formation, and Feixianguan Formation (Figure 2). The strata have gentle production conditions, a stable distribution, and a mild structure, which are conducive to the preservation of shale gas [33]. The well was drilled in the Lower Silurian Longmaxi Formation using a 139.7 mm OD casing (Figure 2). The well has a depth of 4210 m, with a maximum slope angle of 84.25°, and the length of the horizontal section is 1655 m [34].

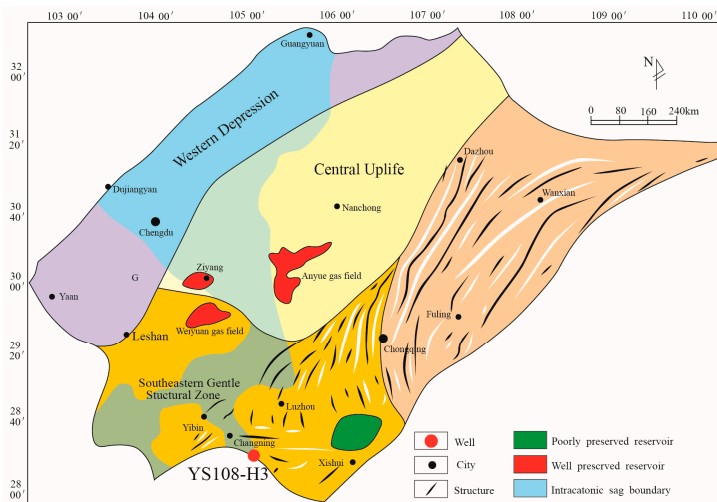

**Figure 1.** Map of the structural units in the YS108-H3 horizontal well group area.

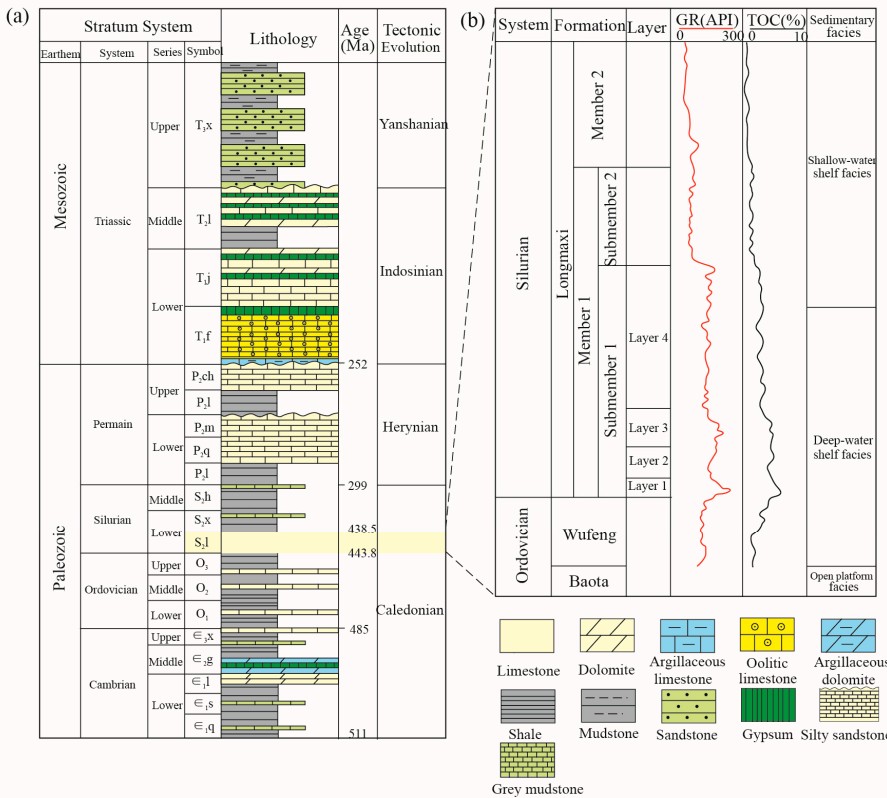

**Figure 2.** Structural diagram of the bottom boundary of the Silurian Longmaxi Formation in the YS108-H3 horizontal well area (the Wufeng Formation is the bottom boundary). (**a**) The stratigraphical map of Paleozoic and Mesozoic in Sichuan basin; (**b**) The stratigraphical map of Ordovician and Silurian in Sichuan basin.

## 3. Materials and Methods

The experimental shale samples were obtained from well YS108-H3 in the Zhaotong shale gas demonstration zone. The basic geological parameters of the shale are listed in Table 1.

**Table 1.** Parameters of calculation model.

| Basic Parameter | Unit | Target Layer | Upper/Lower Layer |
|---|---|---|---|
| Thickness | m | 10 | 20 |
| Elastic modulus | $\times 10^{10}$ Pa | 2.52 | 3.5 |
| Poisson's ratio | / | 0.18 | 0.24 |
| Maximum horizontal stress | $\times 10^{7}$ Pa | 7.1 | 7.1 |
| Minimum horizontal stress | $\times 10^{7}$ Pa | 5.65 | 6 |
| Vertical stress | $\times 10^{7}$ Pa | 6.1 | 6.1 |
| Pore pressure | $\times 10^{7}$ Pa | 2.5 | 2.5 |
| Porosity | % | 0.04 | |
| Fracture toughness | MPa m$^{0.5}$ | 4000 | 8000 |
| Fracture permeability | mD | 0.011 | 0.004 |
| Fracture cohesion | MPa | 6 | 6 |
| Initial fracture aperture | mm | 0.26 | 0.15 |
| Leak-off coefficient | $\times 10^{-12}$ m/s | 1 | 0.5 |
| Fluid density | N/m$^3$ | 9800 | |
| Fracturing fluid viscosity | mPa·s | 18 | |
| Injection time | h | 2 | |
| Injection rate | mL/min | 60 | |

### 3.1. Physical Simulation Experiment

To clarify the propagation behavior and expansion patterns of hydraulic fractures in shale, in this study, hydraulic fracturing model experiments were conducted on samples from the shale reservoirs in the Zhaotong area. The experiment simulated the case of a horizontal well, so the minimum horizontal principal stress was applied along the wellbore direction. By controlling the displacement and keeping experimental parameters such as the fracturing fluid viscosity constant, we observed the extension and propagation of the hydraulic fractures.

The experimental fracturing system consisted of a sample loading frame, wellbore pressure system, confining pressure loading system, and real-time control system (Figure 3). The loading frame had a circular structure, allowing for a maximum rock sample size of 762 mm × 762 mm × 914 mm (length × width × height), effectively reducing the dynamic effects of fracture initiation and the influence of stress boundary effects [35]. The triaxial confining pressure was applied using loading pressure plates placed in the gap between the rock sample and the frame. The fracturing fluid was injected into the pressure plates, causing them to expand and transmit the fracturing fluid pressure to the rock sample's surface, simulating the in situ stress conditions, with a maximum pressure of up to 69 MPa [36]. The wellbore pressure system was controlled by a hydraulic servo system, with a maximum injection rate of 200 cm$^3$/s and a maximum pump pressure at the wellhead of 82 MPa [36]. The real-time control system recorded the wellhead pressure, pump displacement, and triaxial confining pressure data in real time and adjusted the pump rate according to the experimental conditions [37]. Additionally, to visually observe the fracture morphology, red or blue colorants were mixed into the injected fluid [38].

To better observe the extension and propagation of the fractures within the entire wellbore section of the hydraulic fracturing specimen, we cut the post-fracture specimen along the direction parallel to the wellbore. From the cross section of the specimen, it was observed that the fractures traversed the entire specimen (Figure 4). The results of the indoor hydraulic fracturing simulation experiment are presented in Figure 5. To observe the extent of the fracture propagation in the cross section more accurately, the specimen was cut into two parts, as shown in Figure 6.

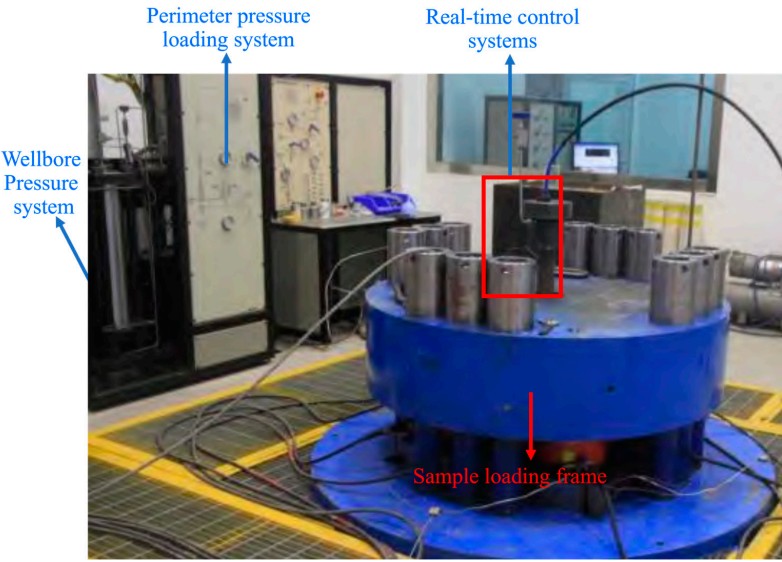

**Figure 3.** Schematic diagram of the test system.

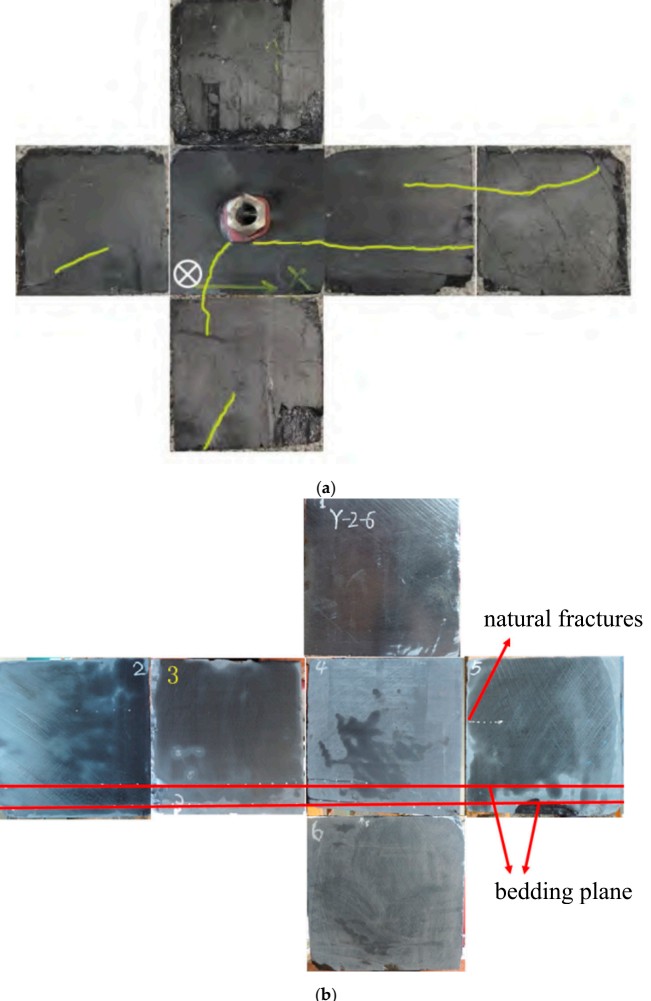

**Figure 4.** Pre-fracturing dissection diagram of the shale sample utilized in the hydraulic fracturing test. (**a**) Specimen from well X1 before hydraulic fracturing; and (**b**) specimen from well X2 before hydraulic fracturing.

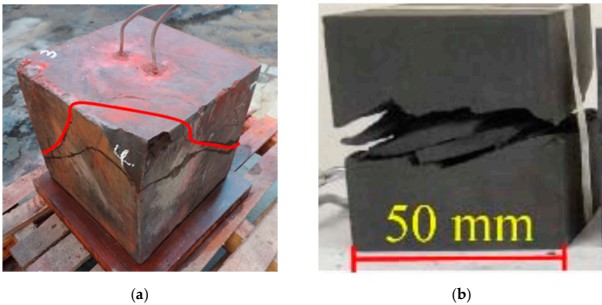

**Figure 5.** Experimental results of hydraulic fracturing of laminated shale specimens. (**a**) Specimen from well X1; and (**b**) specimen from well X2.

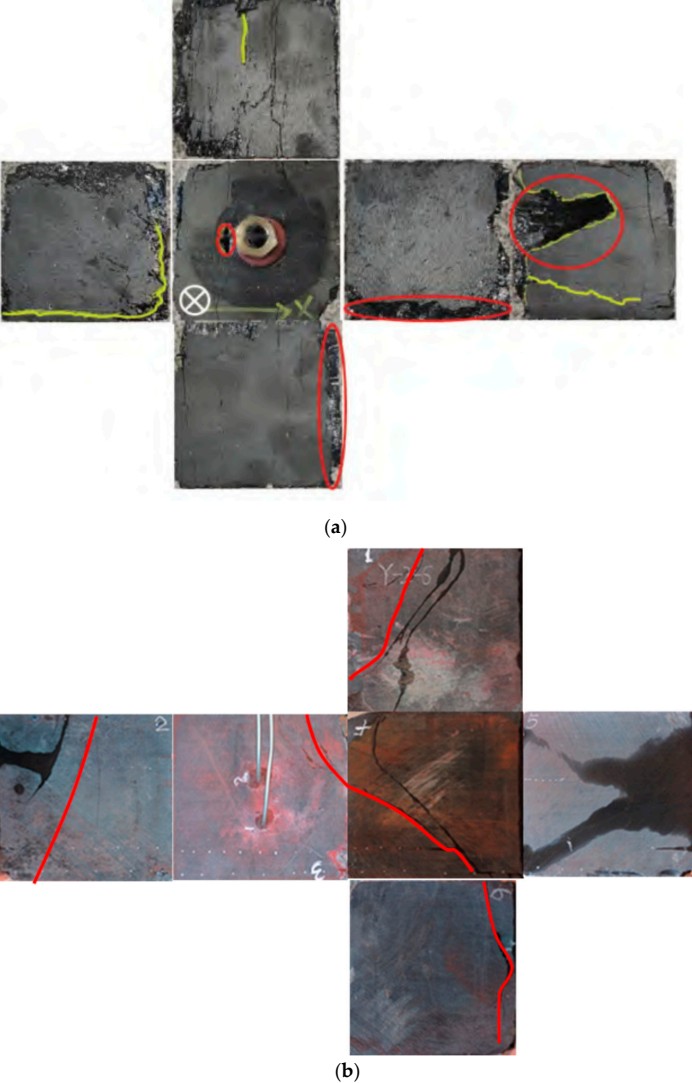

**Figure 6.** Post-fracturing dissection diagram of the shale sample utilized in the hydraulic fracturing test. (**a**) Specimen from well X1; and (**b**) specimen from well X2.

*3.2. Cohesive Element for Simulating Fractures*

During the hydraulic fracturing process, the infiltration of the fracturing fluid into the reservoir rock causes changes in the pore fluid pressure in the vicinity of the fracture and alters the stress distribution within the rock skeleton [39]. This, in turn, leads to rock deformation and damage. The deformation of the rock also affects the flow characteristics of the fluid in the pores and the distribution of the pressure. Therefore, it is necessary

to consider the coupled interaction between the rock damage and deformation processes and the stress field and seepage field within the rock mass when conducting hydraulic fracturing [40].

The ABAQUS software provides a cohesive element for simulating elements that experience failure. In this study, the tensile-separation criterion with element stiffness degradation, which is built into ABAQUS, was employed to simulate the initiation and propagation of fractures. In this study, the ABAQUS 2022 numerical simulation software was used based on the theory of damage mechanics. Cohesive elements were employed to simulate preexisting fractures and investigate the influence of stress differentials on the fracture propagation behavior of hydraulic fractures intersecting with natural fractures under interbed effects [41].

Cohesive elements, also known as cohesive zones, were used to simulate and study the propagation of two-dimensional/three-dimensional fractures by inserting a layer of cohesive elements along the path of the fracture expansion [42]. As shown in Figure 7, although the cohesive elements had a top surface (5, 6, 7, and 8), a middle surface (9, 10, 11, and 12), and a bottom surface (1, 2, 3, and 4), they were considered to be a layer with zero thickness. The damage status of the cohesive elements was determined based on the fracture criteria [43]. When the damage criterion was satisfied, cohesive elements began to experience damage. When the cohesive elements completely failed, they split the middle layer in two, thereby forming a fracture [44].

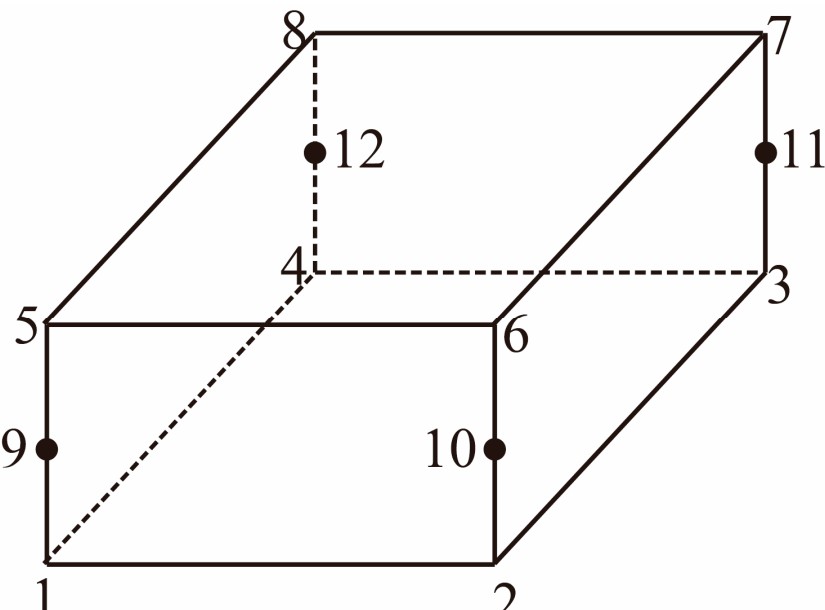

**Figure 7.** Three-dimensional cohesive unit.

### 3.2.1. Fracture Initiation Criteria

The damage model consists of two parts: the initiation criterion and the evolution law. When a material satisfies the damage initiation criterion, it undergoes damage and failure according to the defined evolution law. When the traction force reaches the maximum value that the material can withstand, the damage model exhibits stiffness degradation of the material and structural failure.

The maximum normal stress criterion is adopted as the fracture initiation criterion, which means that, once the stress on any direction of the element reaches its critical value, the element will start to fracture [45].

$$\max\left\{\frac{\sigma_n}{\sigma_n^{max}}, \frac{\tau_s}{\tau_s^{max}}, \frac{\tau_t}{\tau_t^{max}}\right\} = 1, \tag{1}$$

where $\sigma_n^{max}$ is the maximum tensile stress that the element can bear in the vertical direction, which is the tensile strength of coal rock; and $\tau_s^{max}$ and $\tau_t^{max}$ are the maximum shear stress that the element can bear in the two directions, which is the shear strength of coal rock [46].

3.2.2. Fracture Propagation Criteria

The evolution law of the damage describes the rate of the stiffness degradation of the material when it reaches the onset value for damage and failure [47]. The non-dimensional damage factor $D$ is introduced as the criterion for fracture propagation, with a value range of 0–1. When $D = 0$, the material is undamaged, and the material is fully damaged when $D = 1$, so fractures form and continue to propagate. When $0 < D < 1$, the material undergoes damage, and the expression is as follows [48]:

$$\sigma_n = \begin{cases} (1 - D)\sigma_n', \sigma_n' \geq 0, \\ \sigma_n', \sigma_n' < 0, \end{cases} \tag{2}$$

$$\sigma_s = (1 - D)\sigma_s', \tag{3}$$

$$\sigma_t = (1 - D)\sigma_t', \tag{4}$$

where $\sigma_n' \geq 0$ represents the tensile stress that the cohesive element can bear, and $\sigma_n' < 0$ represents the compressive stress that the cohesive element can bear. $\sigma_n, \sigma_s,$ and $\sigma_t$ are the normal stress components and two tangential stress components that the element bears in the current situation; and $\sigma_n', \sigma_s',$ and $\sigma_t'$ are the normal stress component and tangential stress components of the element calculated under linear elastic conditions before material damage occurs [49].

$$\left\{ \frac{G_n}{G_n^c} \right\}^\alpha + \left\{ \frac{G_s}{G_s^c} \right\}^\alpha + \left\{ \frac{G_t}{G_t^c} \right\}^\alpha = 1, \tag{5}$$

where $G_n{}^C$ is the normal fracture energy of the cohesive element (N/mm); $G_s{}^C$ is the first shear fracture energy of the cohesive element (N/mm); $G_t{}^C$ is the second shear fracture energy of the cohesive element (N/mm); and $\alpha$ is the exponent coefficient. If this equation holds, then the total fracture energy of the cohesive element in the mixed mode is $G_C = G_n + G_s + G_t$ [50].

3.2.3. Fluid Equations within Cohesive Elements

After damage, the fluid mainly flows along the tangential and normal directions within the cohesive element.

The tangential flow equation within the cohesive element is as follows [51]:

$$q = -\frac{w^3}{12\mu}\nabla p. \tag{6}$$

The equation for the normal flow along the Cohesive element is as follows [52]:

$$\begin{cases} q_t = c_t\left(p_f - p_t\right), \\ q_b = c_b\left(p_f - p_b\right), \end{cases} \tag{7}$$

where $q$, $q_t$, and $q_b$ are the tangential flow, normal flow on the upper surface, and normal flow on the lower surface of the cohesive element, respectively; $\nabla p$ is the pressure gradient in the length direction of the cohesive element; $\omega$ is the width of the fracture; $\mu$ is the fluid viscosity; $c_t$ and $c_b$ are the filtration coefficients at the upper and lower surfaces, respectively; and $p_t$ and $p_b$ are the pore pressures at the upper and lower surfaces.

*3.3. Three-Dimensional Hydraulic Fracturing Model*

By investigating the relevant information about the reservoir in the study area and obtaining geological data, the basic parameters for the hydraulic fracturing simulation

were determined based on the actual parameters and considering the convergence of the simulation results (Table 1). A three-dimensional hydraulic fracturing model was simulated using software to replicate a thin reservoir and interbedded layers at a depth of 2500 m underground. To simplify the calculations, the model was designed with an intermediate reservoir that was bounded by upper and lower barrier layers, and the wellbore direction was perpendicular to the reservoir and interbedded layers (Figure 8). The model was 60 m high (Z), 50 m wide (Y), and 100 m long (X), and it was partitioned into three layers. The upper and lower two layers acted as barrier layers, each with a thickness of 20 m, and the intermediate layer was the target layer, with a thickness of 10 m. The injection point was located in the center of the X-Y plane, and a cohesive element surface was established along the Z-axis, perpendicular to the X-Y plane, and passing through the injection point. During the simulation, hydraulic fractures propagated along the arranged cohesive element surface, and the geometry of the fractures was analyzed by analyzing the displacement of the cohesive element nodes. To better study the behavior of the fracture propagation between the layers and the changes in the geometry of the fractures, the grid size was reduced near the fracture surface to increase the calculation accuracy while also ensuring the efficiency of the calculations by increasing the grid size in areas far from the fracture surface [53].

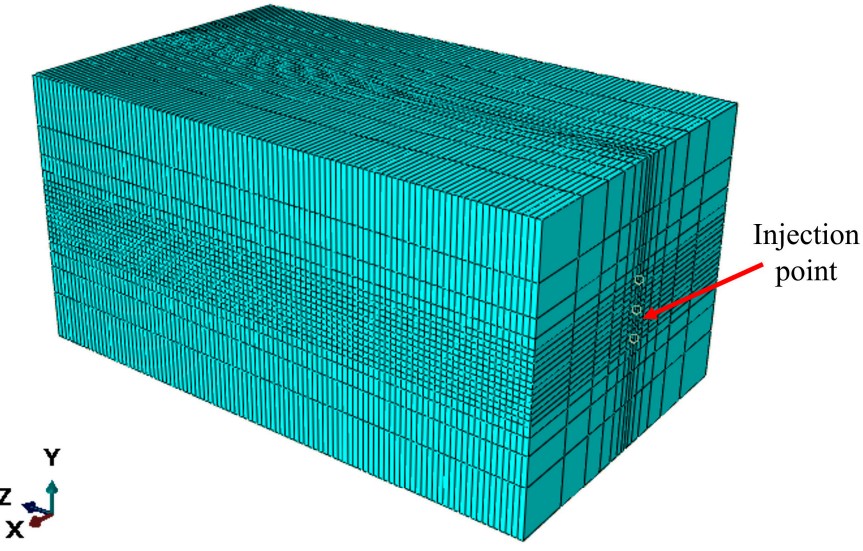

**Figure 8.** Schematic diagram of the hydraulic fracturing model.

The target layer and interbed layer were meshed with C3D8P elements (eight-noded hexahedron, three-directional linear displacement, and three-directional linear poroelasticity). The cohesive elements were modeled with COH3D8P elements (12-noded three-dimensional poroelasticity cohesive element). The mesh was refined in the Y- and X-directions, where the fracture initiated, and the mesh was denser near the initiation point. The Z-direction was uniformly meshed along the fracture length. A total of 28,000 elements were generated in the entire computational model [54].

During the fracturing process, a high-pressure pumping unit injects fracturing fluid at a high displacement and pressure into the shot hole location of the wellbore's target formation. When the fracturing fluid generates a transient holding pressure at the shot hole, which is consistent with the strength of the rock near the wellbore's target formation, as well as the reservoir's geostress field, the rock formations around the wellbore begin to rupture, and the cracks or the already existing natural cracks begin to expand. As the fracturing process proceeds, the fracturing fluid flows forward along the fractures, and the fractured fractures continue to expand. When the length of the fractured fractures reaches the construction requirements, the sand-carrying fluid is pumped into the wellbore, and the next step is to start injecting large quantities of fracturing fluid into the wellbore of the formation.

At this time, the part of the fracturing fluid injected into the wellbore reacts with the original fracturing fluid in the wellbore and the destination layer, and the reaction occurs in the wellbore or the destination layer, thus causing the fracturing fluid to break down and the original part of the fracturing fluid to break down. The viscosity of the original part of the fracturing fluid starts to decrease. In the end, only the proppant is left in the fracture, which is used to prevent the fracture from closing in a later stage, thus forming a connecting channel with a high conductivity, reducing the seepage resistance for oil and gas transport, increasing the flow of oil and gas resources from the far field to the low-pressure zone of the wellbore, and increasing the production of the oil and gas wells. The simulation of the hydraulic fracturing process revealed that the real fractures are characterized by uncertain and extremely complex geometrical rules, which are difficult to express using mathematical formulae. This is an obstacle to the simulation of fracture extension. At present, most of the research conducted in China and abroad has been confined to ideal fractures, i.e., assuming that the fracture is elliptical along the fracture length and fracture height, and the phenomenon of fracture penetration was not taken into account, which is convenient for mathematical expression of the fracture morphology and derivation of the seepage and stress equations. According to the outcrop statistics, core description, and imaging logging interpretation results for the Wufeng–Longmaxi Formation shale in the Sichuan Basin, the natural fractures in the reservoir in well YS108 are mainly laminar and tectonic fractures. The laminar fractures have a smaller inclination angle and higher fracture density than the tectonic fractures, which have a larger inclination angle and lower degree of development. The simulated model represents a two-dimensional vertical cross section of a horizontal well in a shale formation (Figure 9). The objective of this model is to simulate the morphology and characteristics of hydraulic fractures propagating within the plane perpendicular to the rock formation in the horizontal well (Figure 9).

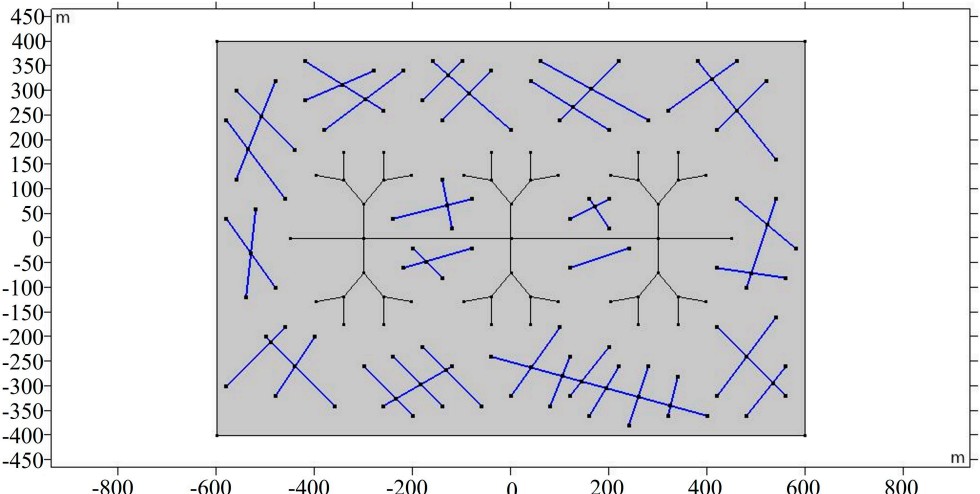

**Figure 9.** Schematic diagram of the natural fracture and hydraulic fracture network.

In order to focus on the morphology and distribution characteristics of the hydraulic fractures in each perforation during the hydraulic fracturing of horizontal well segments, other non-relevant aspects are simplified in this section (Figure 10). The geological and engineering parameters of the model are referenced from Table 1, and local grid refinement is applied to the model (Figure 10).

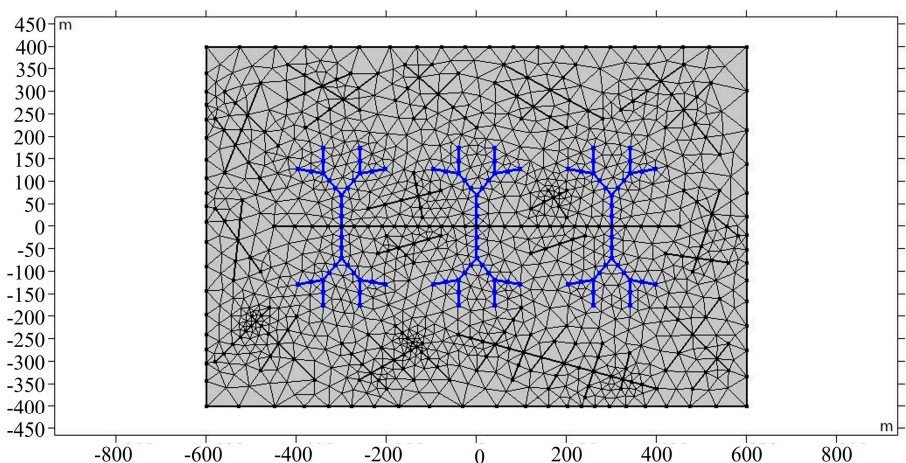

**Figure 10.** Mesh subdivision of the natural fracture and hydraulic fracture network.

## 4. Results and Discussion

Using the hydraulic fracturing finite element model established in this paper, the fracture propagation was simulated for 2 h, and the fracture propagation was analyzed based on the characteristics derived during the calculation process.

### 4.1. Variation in Pore Pressure Inside the Fracture

After the completion of the hydraulic fracturing, the pore pressure distribution cloud map of the fracture revealed a concentrated high-pressure situation within the fracture (Figure 11), whereas it revealed a stepped distribution around the fracture and an elliptical distribution near the fracture tip. This is because the permeability of the reservoir was greater than that of the barrier, and the corresponding flow rate of the fluid that filtered into the reservoir was larger than that of the fluid that flowed into the barrier, resulting in a lower pore pressure on the surface of the barrier fracture than on the surface of the reservoir [55].

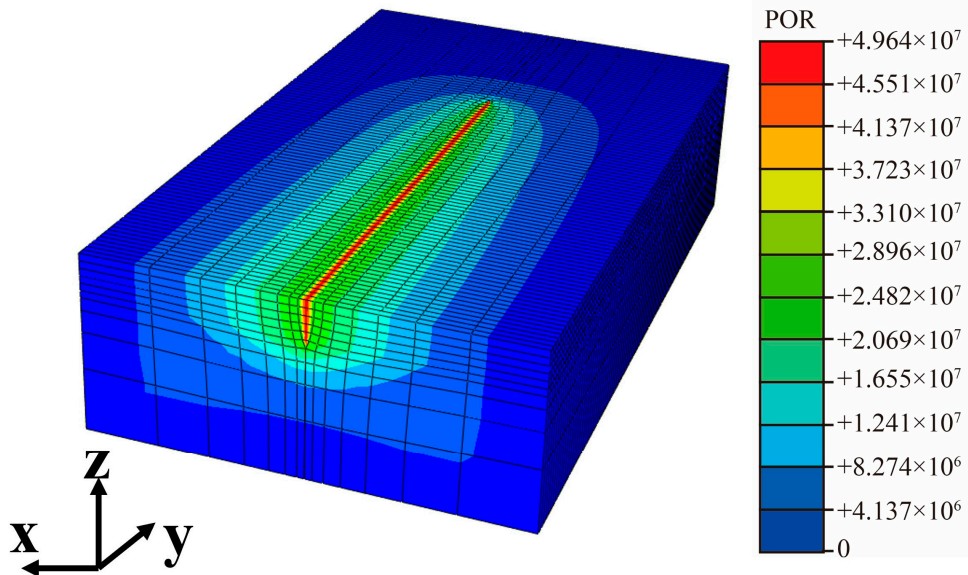

**Figure 11.** Cloud map of the pore pressure distribution in the fracture at the final time.

Figures 12–19 show the pressure variation curve inside the hole during the hydraulic fracturing process at an injection pressure of 80 MPa. As can be seen from Figures 12–19, the pressure inside the central hole gradually increases to a peak from the initial pressure. After the hydraulic fracturing induces the appearance of fractures, the fluid inside the central

hole flows into other pipe domains along the fractures, causing a sudden drop in the water pressure inside the central hole (Figures 12–19). The degree of this sudden drop is closely related to the residual fracture aperture. As the fluid continues to be injected into the central hole and flows into the fractures, small fractures appear around the hole (Figures 12–19). With increasing pressure in the central hole, the fracturing fractures continuously expand towards the boundaries of the model and form connections with the areas of weaker contact with the internal particles, finally extending to the boundary (Figures 12–19). The significant color differences in the different regions where the fractures expand indicate a large difference in velocity (Figures 12–19).

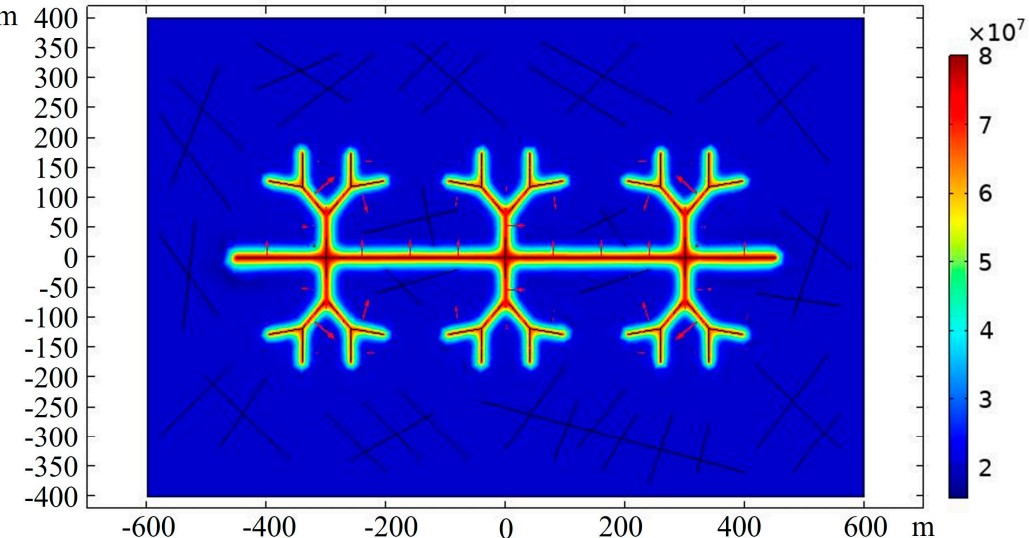

**Figure 12.** Pressure propagation surface maps at 10 min during hydraulic fracturing at 80 MPa.

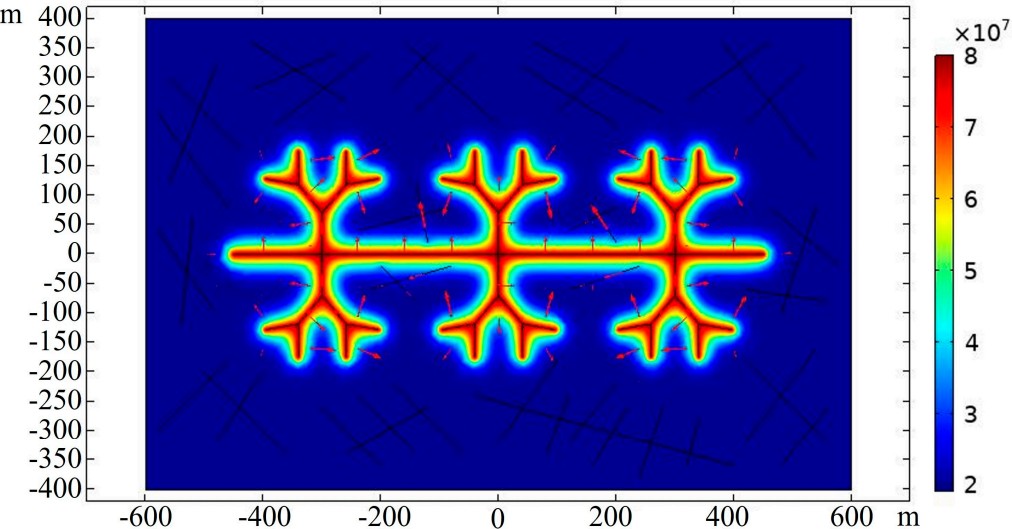

**Figure 13.** Pressure propagation surface maps at 20 min during hydraulic fracturing at 80 MPa.

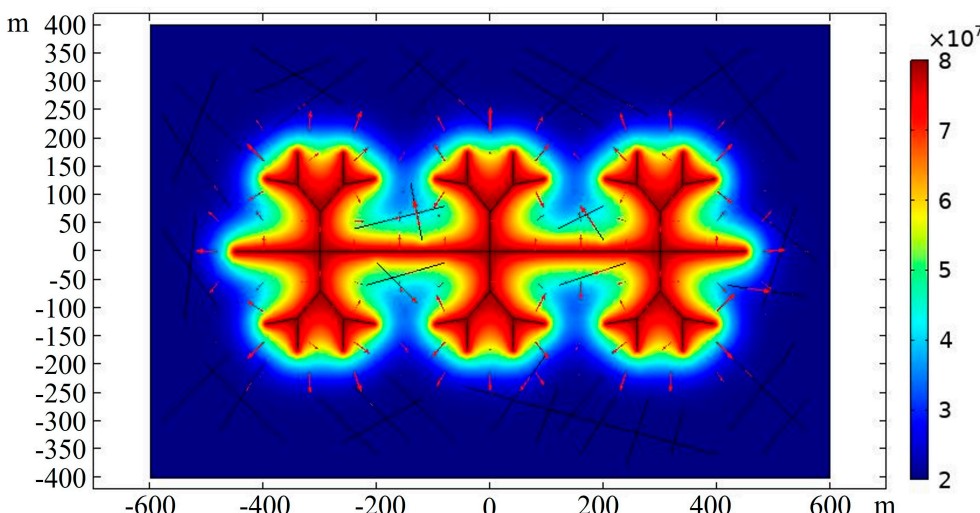

**Figure 14.** Pressure propagation surface maps at 30 min during hydraulic fracturing at 80 MPa.

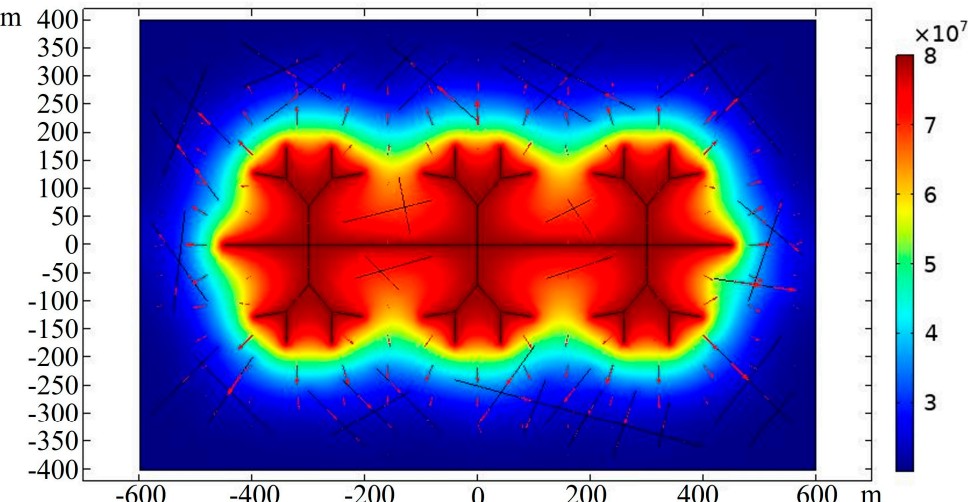

**Figure 15.** Pressure propagation surface maps at 40 min during hydraulic fracturing at 80 MPa.

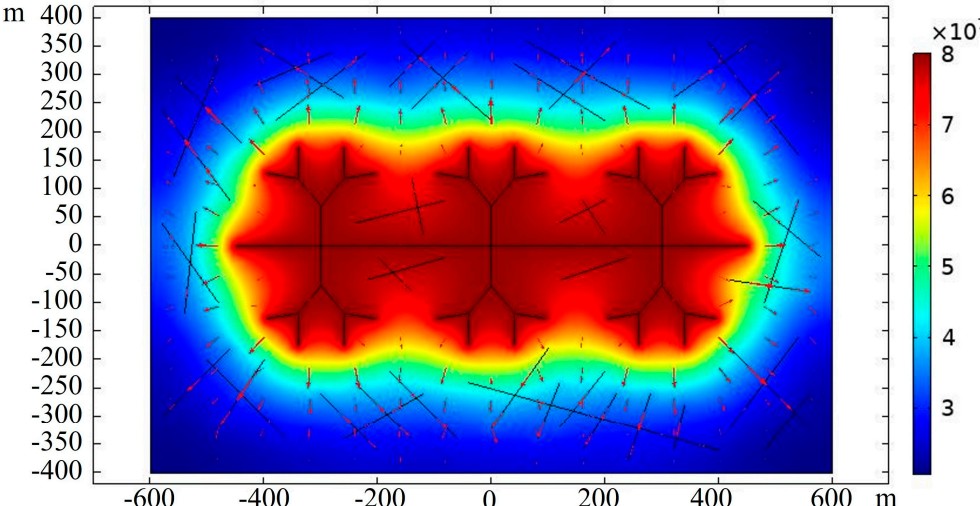

**Figure 16.** Pressure propagation surface maps at 50 min during hydraulic fracturing at 80 MPa.

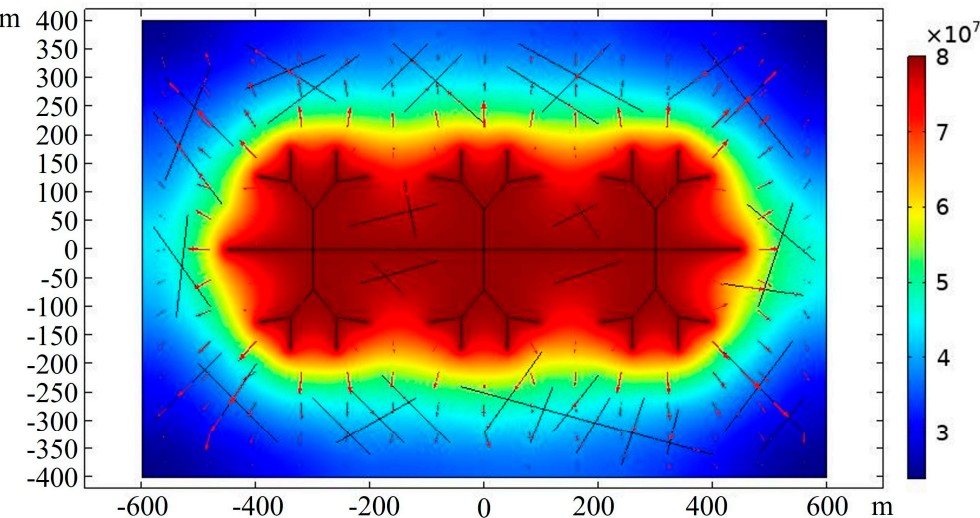

**Figure 17.** Pressure propagation surface maps at 60 min during hydraulic fracturing at 80 MPa.

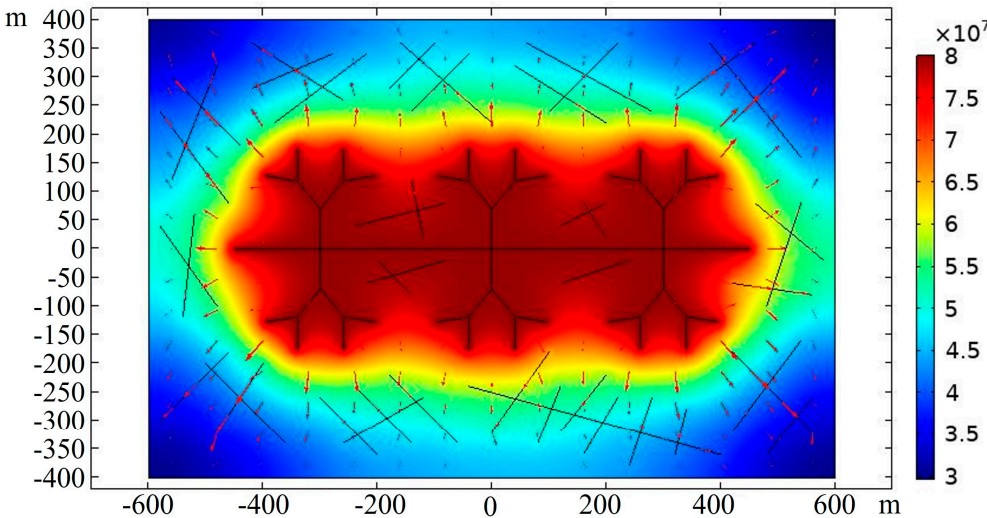

**Figure 18.** Pressure propagation surface maps at 70 min during hydraulic fracturing at 80 MPa.

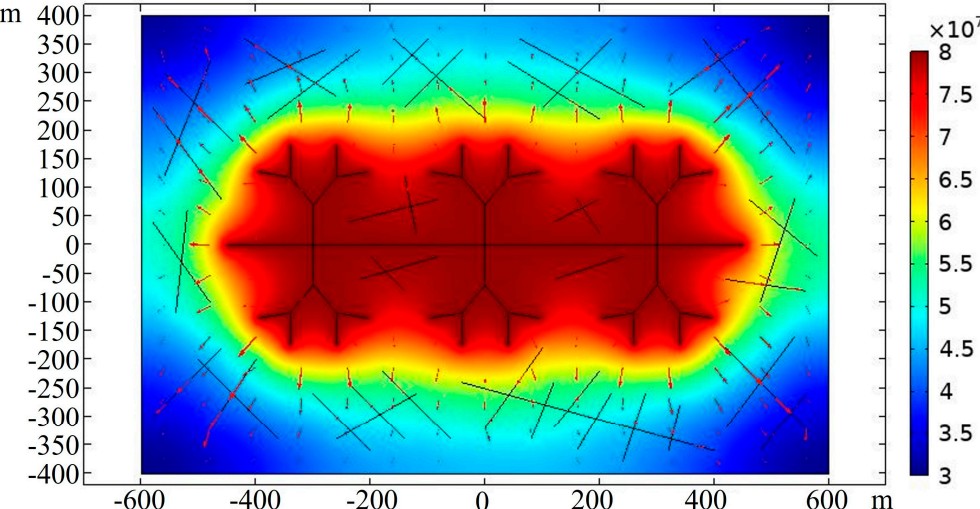

**Figure 19.** Pressure propagation surface maps at 80 min during hydraulic fracturing at 80 MPa.

The injection pressure has a significant impact on the expansion of fractures during shale fracturing. We further considered the hydraulic fracturing process under the influence of different injection stresses at 85 MPa. The resulting hydraulic fracturing fractures under different conditions are shown in Figures 20–26. As can be seen from Figures 20–26, the hydraulic fracturing fractures extend and propagate in the direction of the higher in situ stress. Additionally, by comparing the shapes of the hydraulic fracturing fractures on the different injected pressure, it can be seen that the length of the hydraulic fracturing fractures under 35 MPa is significantly longer than the length of the fractures under 20 MPa (Figures 20–26). This indicates that the difference in in situ stresses has a significant influence on the length of the fracture extension (Figures 20–26). The greater the difference in the in situ stresses, the greater the length of the extension of the hydraulic fracturing fractures (Figures 20–26).

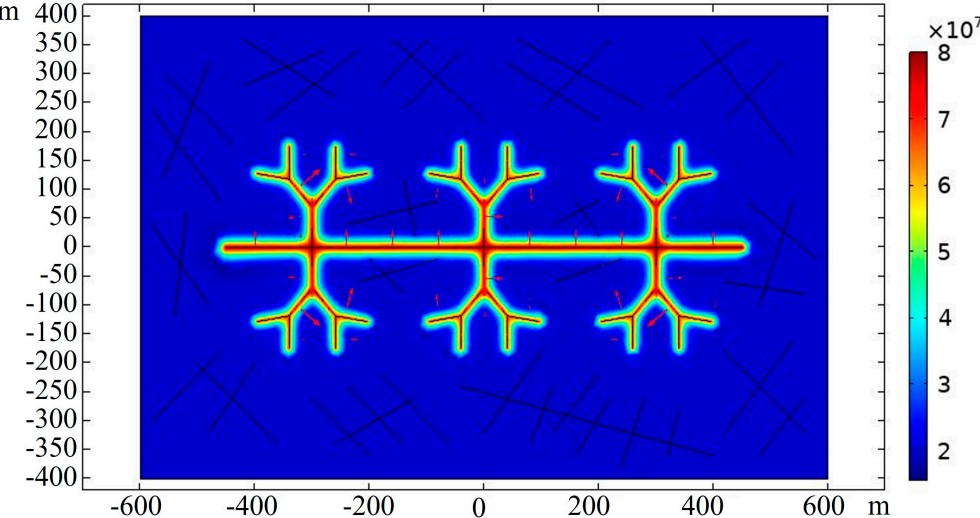

**Figure 20.** Pressure propagation surface maps at 10 min during hydraulic fracturing at 85 MPa.

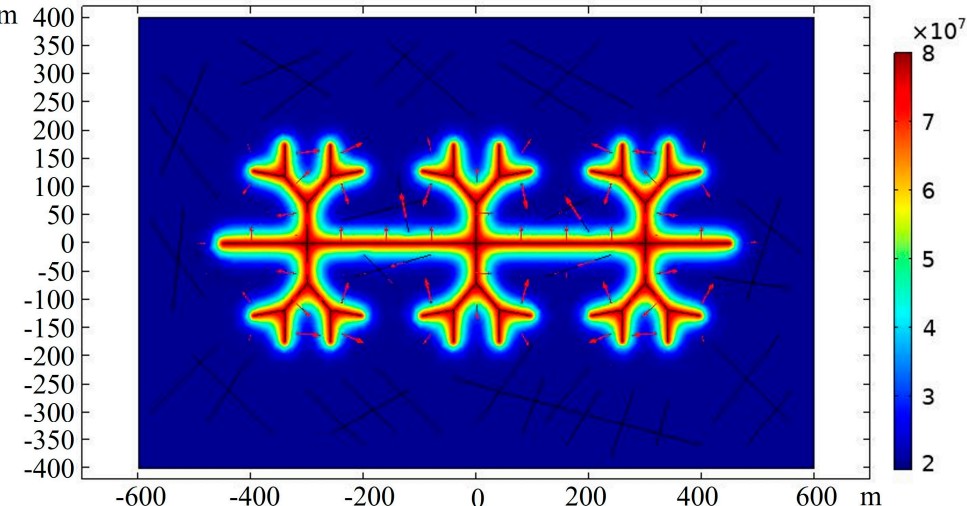

**Figure 21.** Pressure propagation surface maps at 20 min during hydraulic fracturing at 85 MPa.

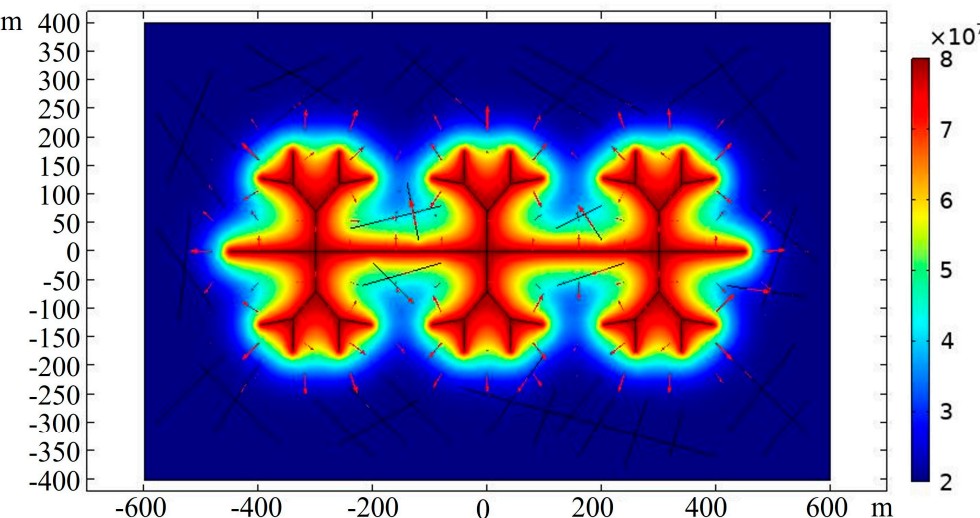

**Figure 22.** Pressure propagation surface maps at 30 min during hydraulic fracturing at 85 MPa.

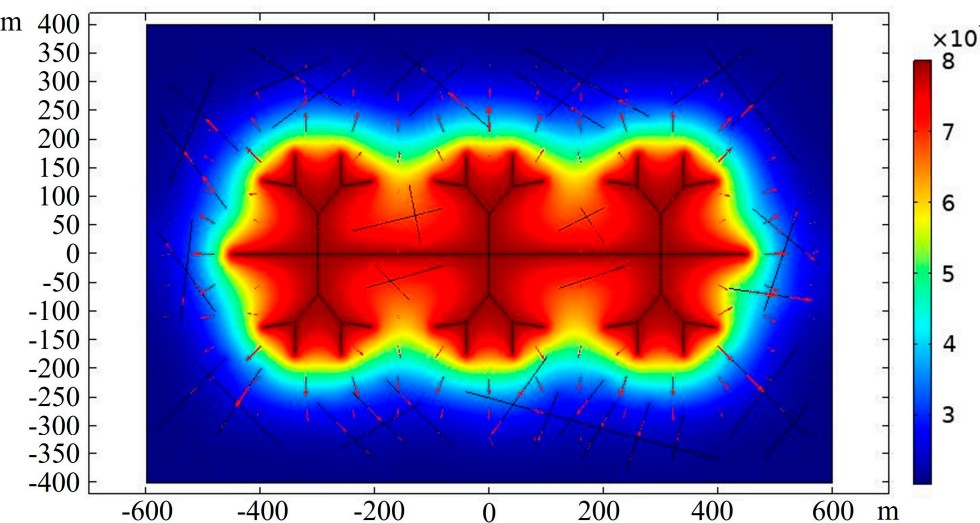

**Figure 23.** Pressure propagation surface maps at 40 min during hydraulic fracturing at 85 MPa.

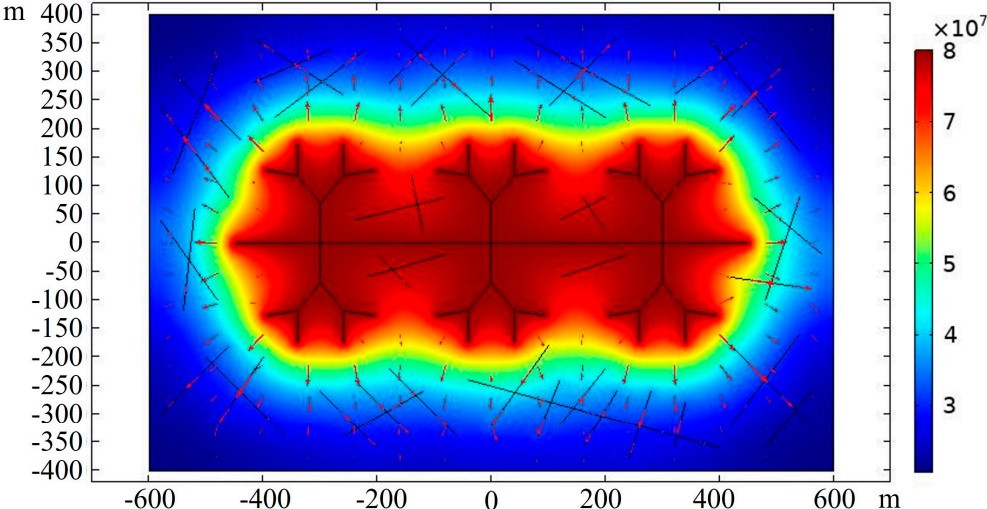

**Figure 24.** Pressure propagation surface maps at 50 min during hydraulic fracturing at 85 MPa.

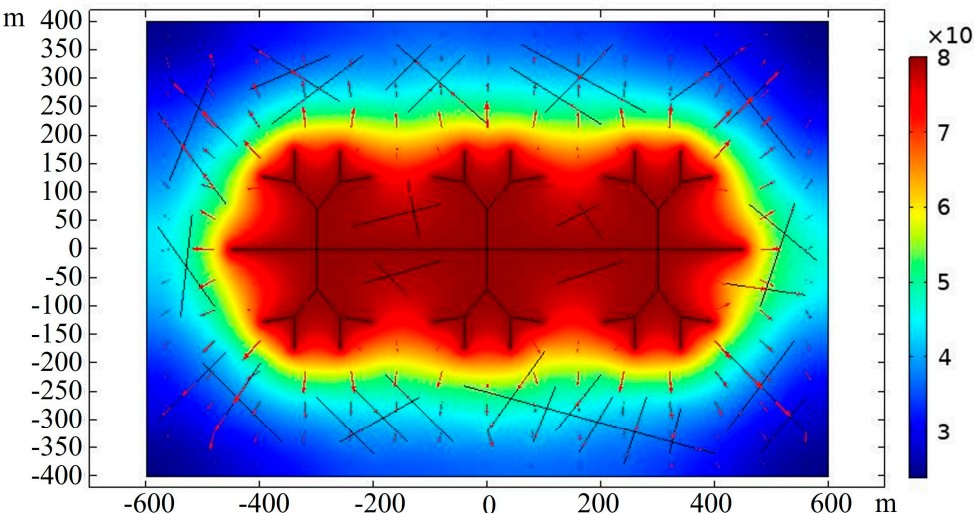

**Figure 25.** Pressure propagation surface maps at 60 min during hydraulic fracturing at 85 MPa.

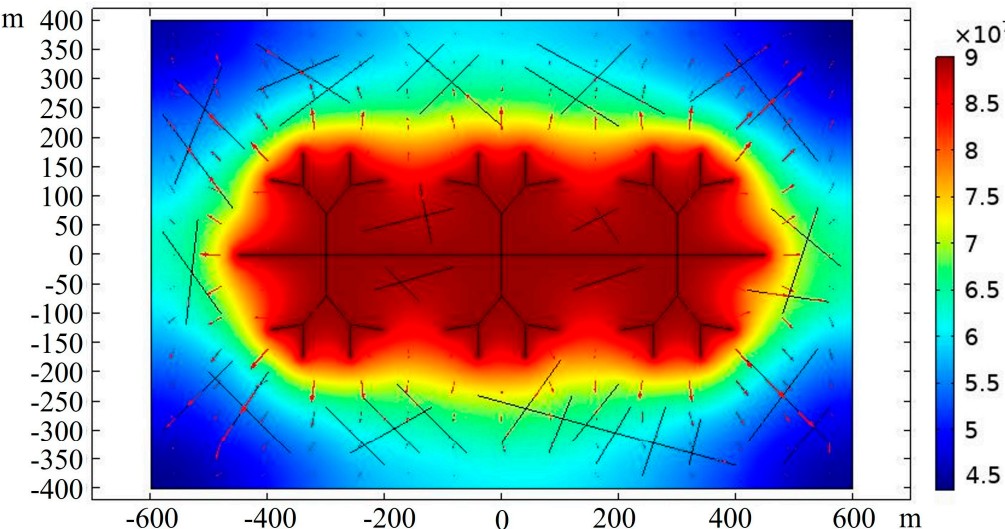

**Figure 26.** Pressure propagation surface maps at 70 min during hydraulic fracturing at 85 MPa.

*4.2. Analysis of Fracture Morphology*

(1)  Variation in fracture geometry

During the initial stage of fracture propagation, before the hydraulic fracture intersects with the natural fractures, the stress difference in the formation has no influence on the fracture propagation behavior. The length of the hydraulic fracture is much larger than its height and width (Figure 11). This is because as the hydraulic fracture penetrates deeper into the formation, it encounters greater resistance, resulting in pressure concentration inside the fracture. Moreover, due to the obstruction of the upper and lower layers, it is difficult for the fracture height to increase, so the length of the fracture is much greater than its height [56].

To observe the expansion morphology and law of the fractures in the formation, in Figure 27, the display of the other elements is hidden and only the layer of the cohesive elements forming the fracture is displayed.

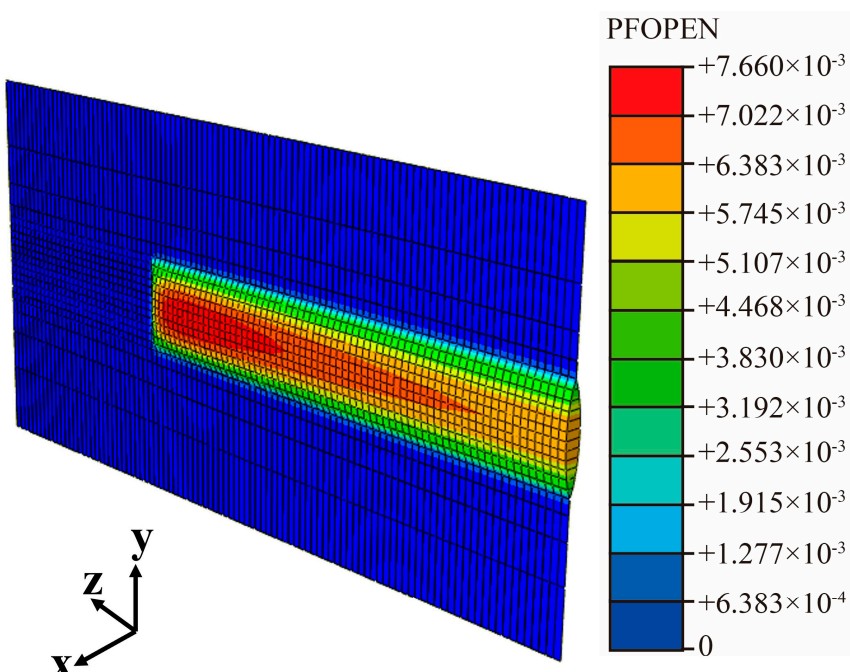

**Figure 27.** Fracture morphology at the end of hydraulic fracturing.

The results show that the fracture aperture is wider in the middle and narrower on both sides, indicating that the fracture width gradually decreases symmetrically from the center toward both sides in the vertical section of the fracture width [57]. When the injection point is in the reservoir, the fracturing fluid first enters the reservoir and then infiltrates into the barrier layer. Due to the difference in permeability, the volume of the fracturing fluid entering the reservoir is larger than that entering the barrier layer, and the stress of the barrier layer is greater than that of the reservoir. Consequently, the resistance to fracture initiation increases, resulting in a larger fracture width in the middle and smaller widths on both sides [58]. Along the length direction of the fracture, the aperture gradually decreases to closure at the fracture tip, and the width of the fracture is smaller near the wellbore than at the far end [59]. This is because as the hydraulic fracture extends to a certain range, the filtration loss in the fracture increases, which may cause a slight decrease in the fracture width, resulting in a smaller fracture width in the area near the wellbore than at the far end.

According to the hydraulic fracturing numerical simulation results, it is concluded that the maximum length and maximum width of the formed fractures gradually increase with increasing injection pressure (Figures 28–30). This indicates that if the injection pressure of the fracturing fluid during hydraulic fracturing can exceed the rock fracturing pressure of the shale, then continuing to inject a large amount of fracturing fluid during the fracturing process will cause the fractures to continuously propagate and expand, and the area affected by the fractures will gradually increase (Figures 28–30). In the later stages of hydraulic fracturing, the rate of increase in the fracture width is slightly greater than the rate of increase in the fracture length (Figure 28). This phenomenon indicates that during hydraulic fracturing, after the injection pressure continuously increases to a certain value, the ability of the injected fracturing fluid to create longer fractures in the later stages of hydraulic fracturing is lower than its ability to create wider fractures [60]. In addition, as the measurement point for the maximum fracture width is at the initiation point of the fracture, which is the tip of the perforation interval, and due to the accumulation of fracturing fluid at the bottom of the well, the pressure at this point is often the highest. Therefore, the width of the fracture at this location is often much larger than the average width of the fracture, leading to prediction errors.

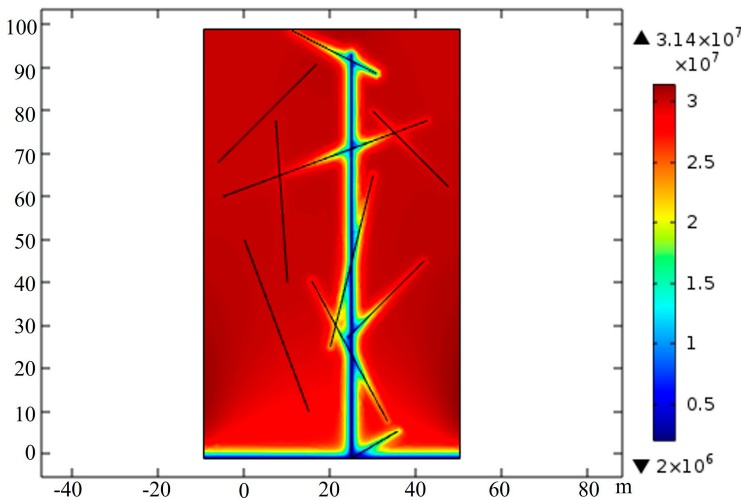

**Figure 28.** Variations in fracture distribution on 10 min during hydraulic fracturing at 85 MPa.

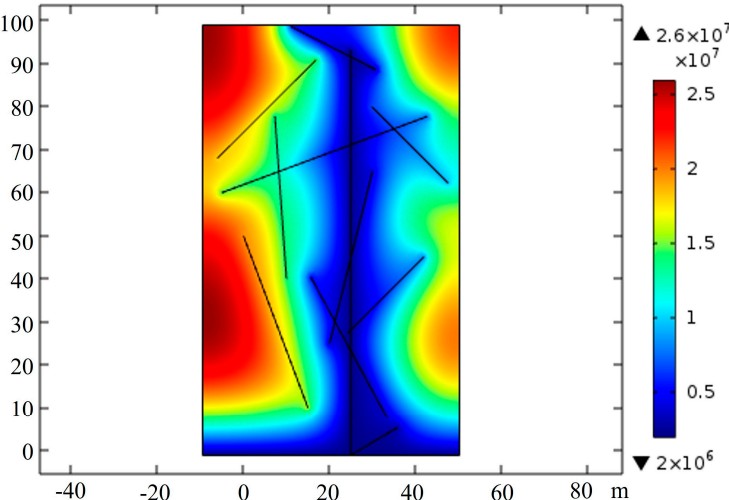

**Figure 29.** Variations in fracture distribution on 50 min during hydraulic fracturing at 85 MPa.

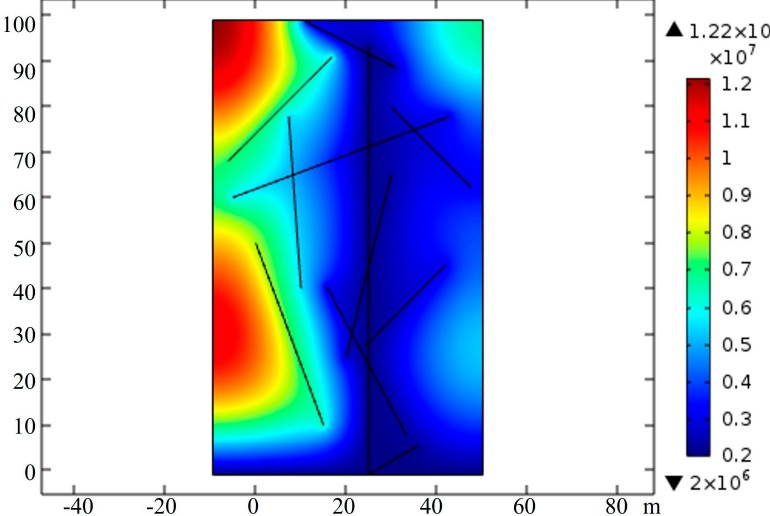

**Figure 30.** Variations in fracture distribution on 70 min during hydraulic fracturing at 85 MPa.

When the fracturing fluid is injected at a pressure of 85 MPa, it can be observed that, on the 10 min, the fractures exhibit significant expansion and a gradual increase in their angles (Figure 31). After 50 min of fluid injection, the fractures rapidly propagate and extend within the interlayers, with larger opening angles than those on the first day (Figure 32). As the injection time increases to 70 min, the fractures in the interlayers continue to extend and expand (Figure 33).

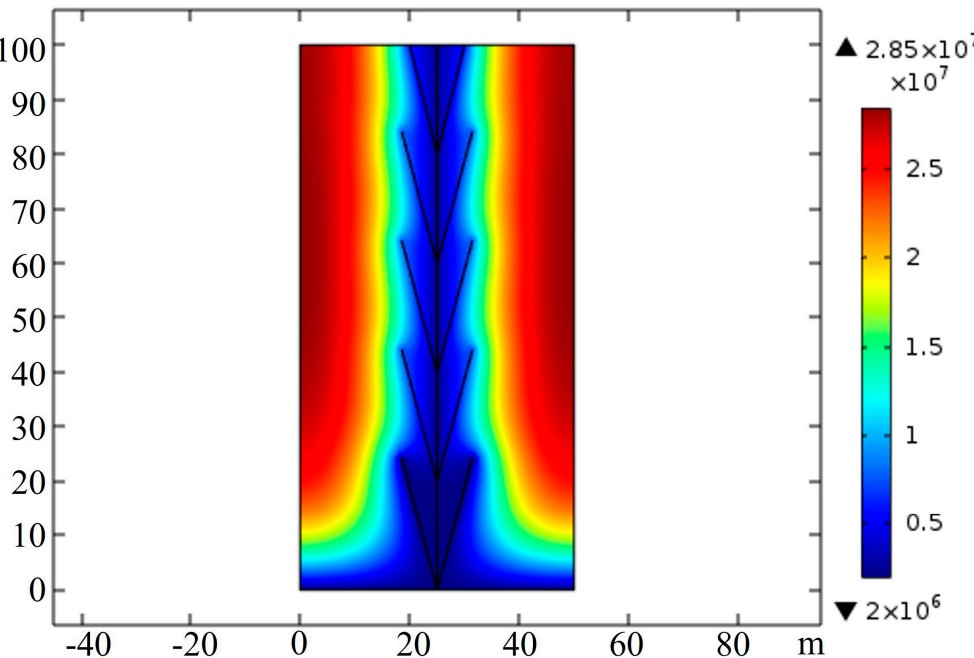

**Figure 31.** Changes in fracture angles on 10 min during hydraulic fracturing at 85 MPa.

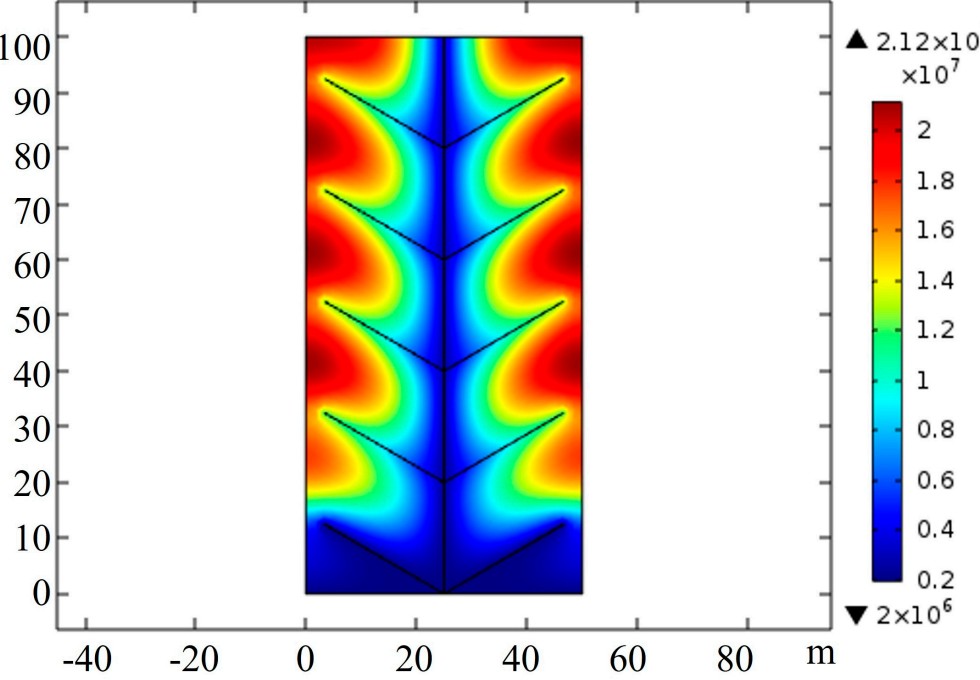

**Figure 32.** Changes in fracture angles on 50 min during hydraulic fracturing at 85 MPa.

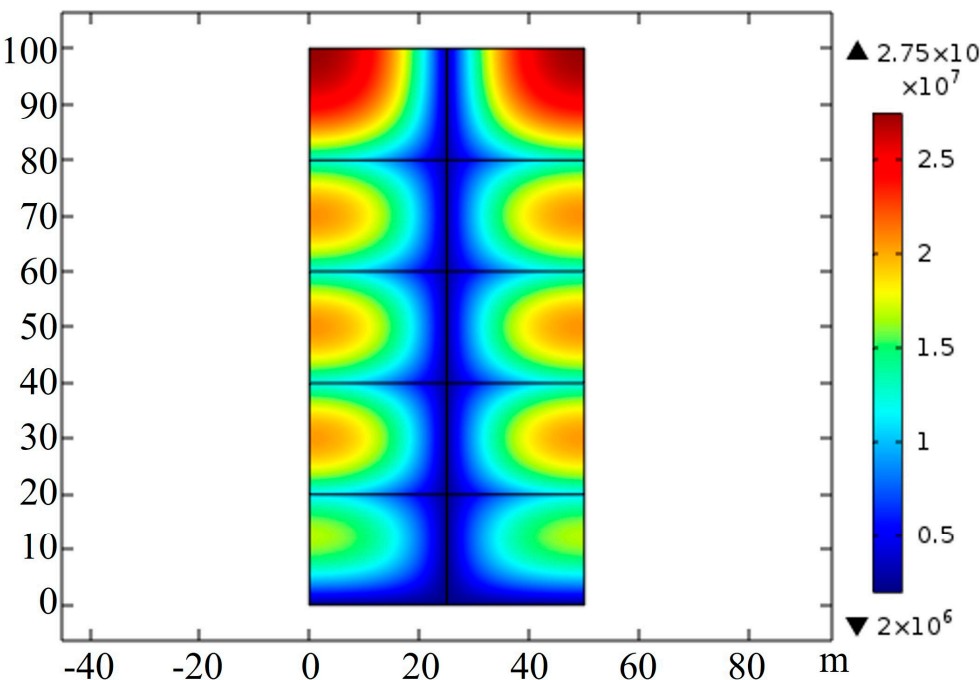

**Figure 33.** Changes in fracture angles on 70 min during hydraulic fracturing at 85 MPa.

(2)   Variations in pore pressure and fracture width at the injection point with time

The characteristics of the variations in the pore pressure and fracture width with time at the injection point (Figure 34) indicate that the pore pressure at the injection point initially increases sharply, then decreases, and finally stabilizes over time. During the first 2 s of injection, the pore pressure increases rapidly to 76 MPa, and then it drops sharply, indicating that the fracture initiation pressure is 76 MPa. After a period of fluctuation, the pore pressure stabilizes at around 48 MPa. The fracture width at the injection point increases gradually in a curved pattern, starting with small fluctuations after the fracture initiation and reaching the maximum width, before gradually decreasing and finally stabilizing.

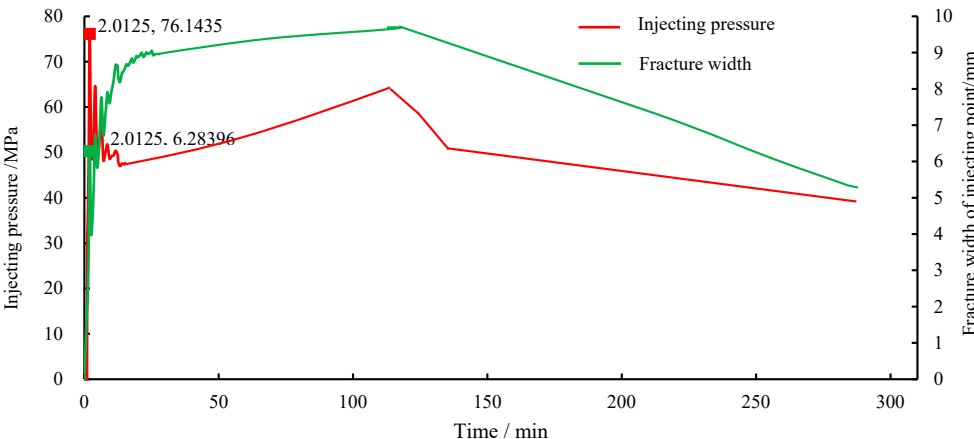

**Figure 34.** Variations in injection point pressure and fracture width with time.

At the beginning of the injection of the fracturing fluid into the reservoir, a buildup of pressure near the injection point can easily lead to an instant increase in the fracture, and, once initiated, the fracture rapidly propagates along the predetermined direction of the elements. Under the effect of the horizontal stresses, the injection pressure decreases, and the fracture width briefly decreases. With the gradual replenishment of the injection fluid, the fracture width gradually increases and exhibits a step-like propagation pattern before

reaching the maximum width. After reaching the maximum width, the loss of fluid in the fracturing fracture increases, resulting in a slight reduction in the fracture width, and then it then gradually stabilizes.

(3)    Relationship between fracture width and height/length changes

Taking the injection point as the research object, the width curves of the fractures at different simulation times along the trajectory passing through the injection point and the fracture height were plotted (Figure 35). The results show that the fracture width at the injection point reaches the maximum value of 9.05 mm at the simulation time of 26 s, and then it gradually decreases until the fracture width at the injection point drops to 6.33 mm at the final simulation time. This indicates that the fracture width increases rapidly in the early stage of opening and begins to decrease slowly after reaching the maximum value. This is consistent with the analysis of Figure 34, i.e., the loss of the hydraulic fracture increases when the fracture expands to a certain range, leading to a slight decrease in its width.

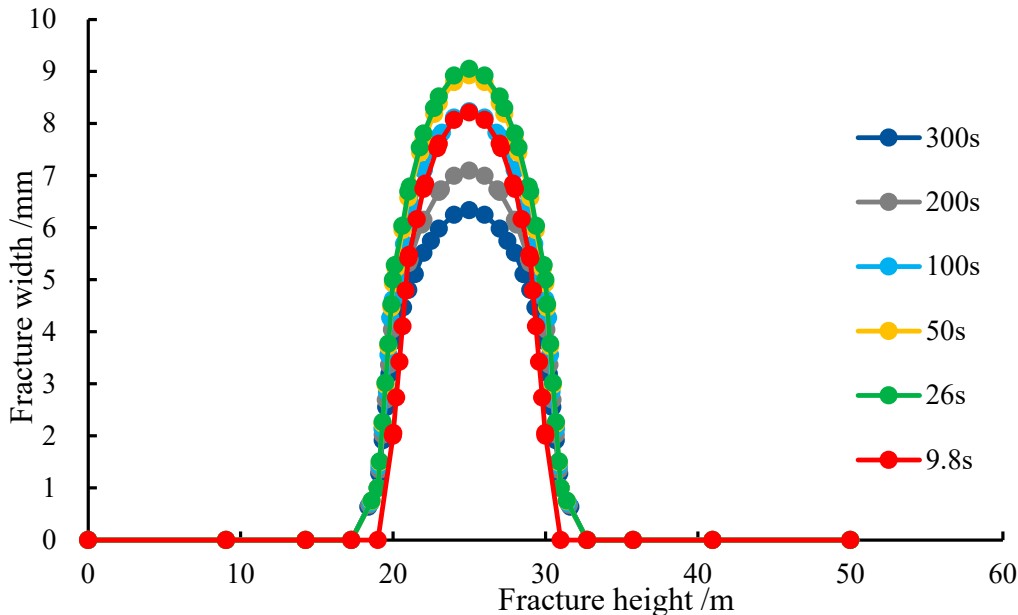

**Figure 35.** Plot of fracture height vs. fracture width.

The fracture width along the fracture length at different simulation times was plotted with the injection point as the starting point (Figure 36). The results show that the fracture width increases rapidly in the initial stage, and both the length and width of the fracture increase with the continued injection of the fracturing fluid. When the fracture width increases to a certain degree, the growth rate decreases. After reaching the maximum value (9.05 mm) at the injection point, the fracture width gradually decreases and slightly increases with increasing distance from the injection point. The sudden drop phenomenon occurs at the end of the fracture width curve (Figure 36), which suggests that there is a critical fracture width in the middle section of the fracture during hydraulic fracturing [14]. During the hydraulic fracturing process, when the fracture width is smaller than a certain critical value, the fracture can only be considered to be a pore flow channel; in contrast, when the fracture width is larger than the critical value, it can form a high-conductivity fracture, which is expected in the fracturing process. Based on Figure 34, when the pore pressure at the injection point first enters the stable stage after a sudden drop, the corresponding fracture width can be approximately regarded as the critical fracture width that occurs during the fracture propagation process. Therefore, the critical width of the reservoir fracture in this model is 8.18 mm.

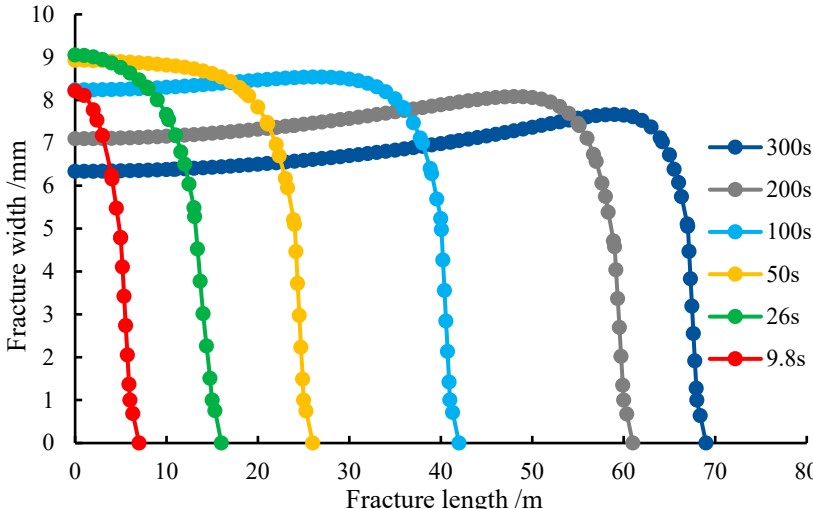

**Figure 36.** Plot of fracture length vs. fracture width.

### 4.3. Analysis of Fracture Breakthrough in Upper and Lower Barriers

During the early stage of hydraulic fracturing, due to the difference in the formation stresses and elastic moduli, the fractures initially propagate in the target layer. From 0 to 8 s, both the fracture length and height change, and the height remains unchanged from 8 to 10 s, whereas the length changes. At 10 s, the fracture breaks through the upper and lower barriers (Figure 37a), and both the fracture height and length change from 10 to 13 s. From 13 to 300 s, the fracture height remains unchanged, whereas the length continues to change. As the hydraulic fracturing fluid continues to be injected, the pore pressure continues to increase. Although there is not a significant difference in formation stresses and elastic moduli between the target and barrier formations, when the pore pressure increases sufficiently to overcome the resistance to fracture extension in the vertical direction, the fracture breaks through the upper and lower barriers.

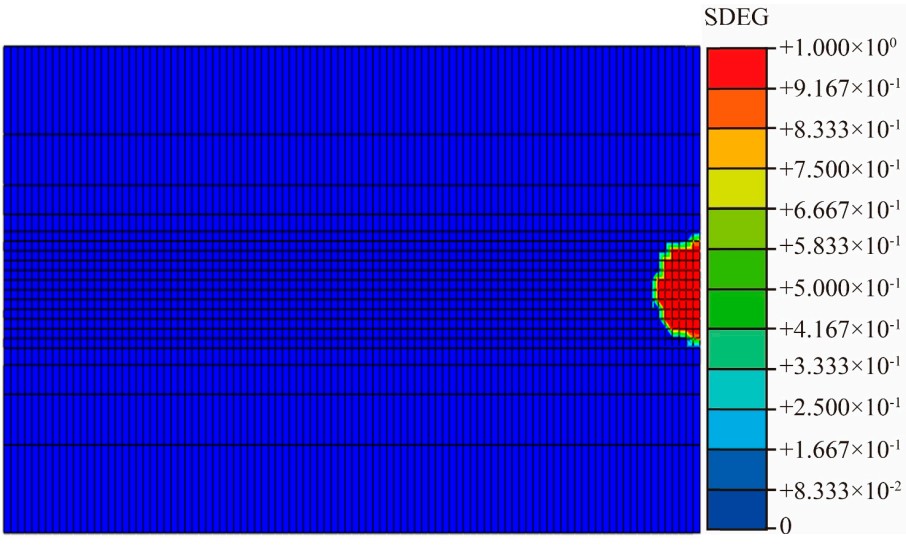

(**a**)

**Figure 37.** *Cont.*

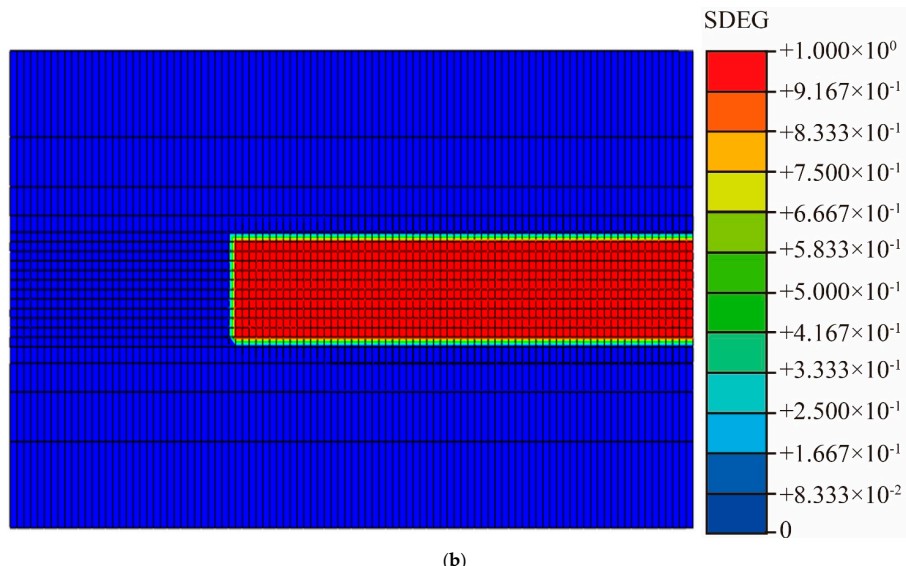

(**b**)

**Figure 37.** Diagram of the fracture breaking through the barrier. (**a**) The fracture breaks through the upper and lower barriers at 10 s during hydraulic fracturing; and (**b**) fracture morphology at the end of fracturing.

The differences in the physical properties of the layers affect the extension range of the hydraulic fracture network, and the rock permeability affects the pressure transmission efficiency of the fracturing fluid within the fracture. In shale reservoirs, dense shale is developed, and the extremely low permeability causes significant resistance to fluid flow. In coal rocks, the face cleat and butt cleat intersect to form an effective flow channel. The large difference in the interlayer permeability limits the extension ability of the fracture height. The differential stress at the barrier mainly controls the expansion of the fracture. The minimum horizontal principal stress difference between the layers and the interface properties are the main factors affecting whether the hydraulic fracture can penetrate the rock interface. When the minimum horizontal principal stress difference is less than 4 MPa, the hydraulic fracture penetrates the rock interface and forms barriers or cross fractures. When the minimum horizontal principal stress difference is greater than 4 MPa, the hydraulic fracture expands to the rock interface, and the fracture turns to extend along the interface, forming T-shaped or blunt fractures. The hydraulic fracture cannot communicate with the adjacent layers [15]. The minimum horizontal principal stress difference of the reservoir is 3.5 MPa, which is less than 4 MPa. The hydraulic fracture penetrates the rock interface and breaks through the upper and lower barriers, which is unfavorable for fracture height control and improvement of the fracturing effect.

*4.4. Simulation Experiment of Hydraulic Fracturing of Shale Specimens*

By observing the fractured specimens, it was found that the hydraulic fracturing fluid flowed out of the two faces in the direction of the applied vertical stress (Figure 35). It can be inferred from this observation that the fractures propagate and extend outward in a bimodal pattern. Therefore, it is expected that secondary fractures have formed in the seam, and they not only propagate longitudinally but also extend continuously in the horizontal direction within the seam.

As the specimen was cut along the direction perpendicular to the wellbore, we could observe the fracture surfaces in the cut cross section (Figure 6). On the cross section, we observed large patches of remaining green tracer, indicating that the fractures successfully initiated and propagated during the fracturing process, and the fracturing fluid carrying the tracer continuously flowed across the fracture surfaces. The significant amount of green tracer within the interbed also suggests that the fractures not only traversed the interbed longitudinally but also extended horizontally within the interbed. Due to the high

propensity for fracturing and the weak development of joints in the interbed compared to conventional rock formations, we observed that the fracture surfaces within the interbed were more fragmented than in the reservoir. These observations indicate that under the given experimental conditions, the interbed does not impede the vertical propagation of the hydraulic fractures and cannot form an effective barrier zone. The fractures can extend into and penetrate the interbed. The interbed does not provide a barrier to the vertical propagation of hydraulic fractures in the shale reservoir, and its ability to control the fracture height is relatively poor. Fractures can enter the interbed and continue to propagate and expand within it.

Due to the indoor nature of the experiment, only vertically oriented wells with certain boundary conditions were simulated, and the simulated geological background corresponded to a single point within the formation. The main observations focused on the initiation time, initiation pressure, and localized propagation of the fractures around the hydraulic fractures. Therefore, in the moment that the fracture initiation limit was reached, the shale specimens experienced almost simultaneous initiation and propagation of cracks, with a duration of only 1 to 2 s.

In field hydraulic fracturing experiments, the propagation of fractures takes a certain amount of time, and the evolution of the fracture propagation can be divided into multiple stages based on the changes in the pressure and temperature over time. In contrast, indoor experiments involve instantaneous initiations of fractures in shale specimens, and the initiation and completion of the fracture propagation occur almost synchronously. Although the specimen size is small, it facilitates the observation of the changes that occur at the moment of fracture initiation in the shale specimens.

Analysis of the morphology of the fractures on the lower surfaces of the shale specimens before and after hydraulic fracturing revealed that the fracturing fluid can induce fractures and enhance the permeability of shale reservoirs. The variations in the injection volume of the fracturing fluid affected the morphology of the fractures on the lower surfaces of the shale specimens, suggesting that altering the injection volume of the fracturing fluid may modify the width of the fractures. The morphology of the fractures on the lower surfaces revealed that the extension of fractures occurs along the direction perpendicular to the minimum principal stress, which is attributed to the experimental conditions, i.e., the confining pressure was lower than the axial stress. In most cases, hydraulic fractures tend to propagate in a direction perpendicular to the plane of the minimum principal stress.

To visualize the expansion of the fractures on the surface of the shale specimens, two side surfaces that exhibited continuous fracture expansions from the lower surface of the specimen were selected for detailed depiction of the surface fractures after the hydraulic fracturing (Figure 38). The primary fractures on all of the side surfaces extended along the direction of the maximum axial stress. However, due to the presence of natural bedding in the shale and the existence of micro-pores within the specimens, the fractures deviated from the central axis when propagating from the interior to the surface (Figure 38). Some of the fractures had orientations perpendicular to the central fracture due to the heterogeneous and anisotropic nature of the actual shale specimens. Increasing the injection volume of the hydraulic fracturing fluid led to narrower fracture widths (Figure 38).

Analysis of the morphologies of the fractures on the side surfaces of the shale specimens under the influence of different injection volumes of the fracturing fluid revealed that altering the injection volume affects the width of the primary fractures. As the injection volume of the fracturing fluid increased, the width of the primary fractures in the shale specimens gradually decreased.

To investigate the influences of different injection volumes of fracturing fluid on the widths of the primary fractures in the shale, quantitative analysis of the widths of the primary fractures on the specimen surfaces was conducted. The results are shown in Figure 39, which presents a comparison of the widths of the primary fractures after hydraulic fracturing. Figure 39a provides a magnified view of the morphology of the primary fractures on the surfaces of the shale specimens. For ease of comparison, each

image is annotated according to a certain scale. After hydraulic fracturing, the width of the primary fractures on the surface of specimen 1-1 was within the range of 0.382 to 0.802 mm, with maximum fracture widths of 0.802 mm and 0.239 mm, i.e., a decrease of 70.19% in the maximum fracture width. This indicates that as the injection volume of the hydraulic fracturing fluid increased, the range of the primary fracture width on the surface of the shale specimens decreased.

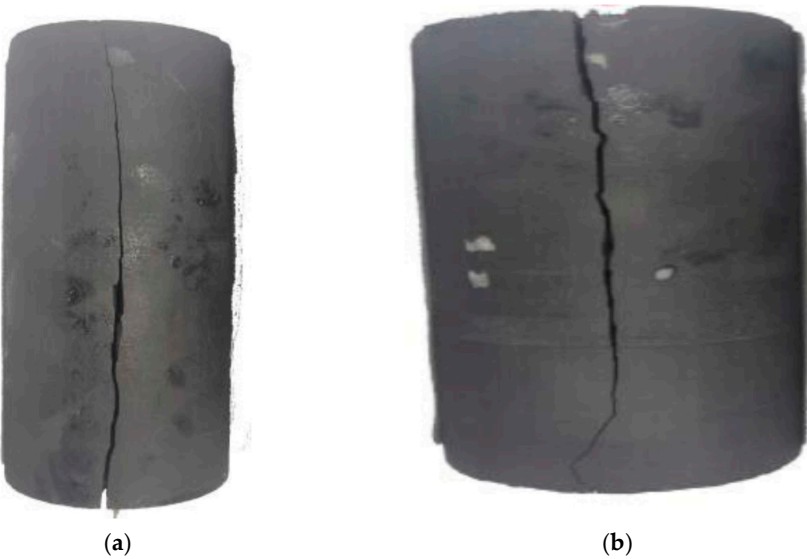

(**a**)                    (**b**)

**Figure 38.** Changes in the lateral surface morphology of the shale after the hydraulic fracturing test. (**a**) The specimen from well X1; and (**b**) the specimen from well X2.

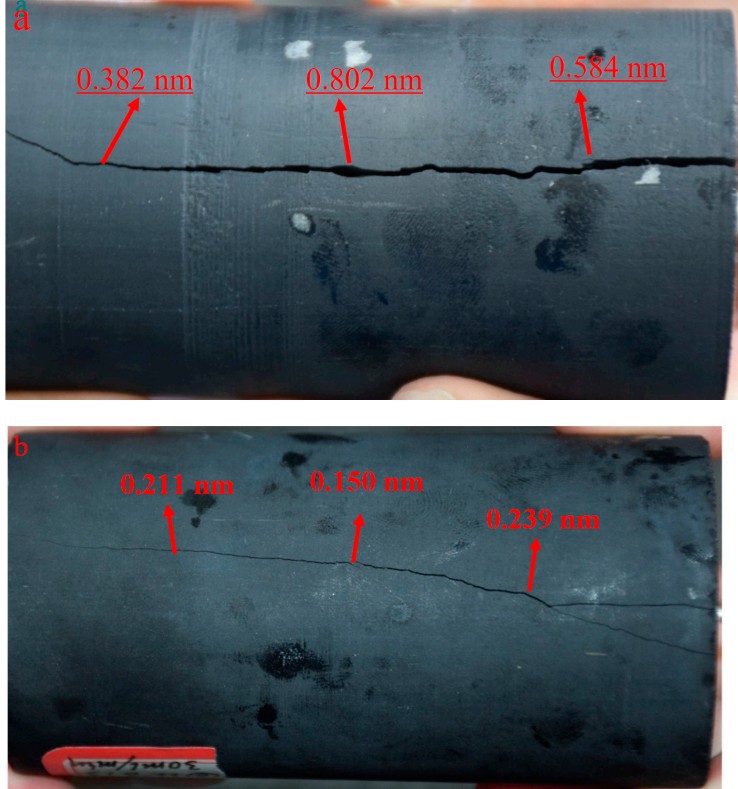

**Figure 39.** Width of main fractures on the surface of the shale specimens. (**a**) The specimen from well X1; and (**b**) the specimen from well X2.

Analysis of the morphological changes in the fractures on the side surfaces of the shale specimens before and after hydraulic fracturing revealed that conducting hydraulic fracturing experiments can generate multiple primary fractures and induce secondary fractures that extend along the bedding planes. The fracture network on the side surfaces exhibited the highest complexity.

Through comprehensive analysis of the indoor hydraulic fracturing experiment and finite element numerical simulation results, it was found that they were consistent with each other. The hydraulic fractures extended from the shale into the interbed and continued to expand within the interbed. Therefore, when conducting hydraulic fracturing operations in reservoirs containing coal interbeds, it is necessary to optimize the construction plan and take certain measures to control the vertical expansion of the fractures as much as possible, confining the fractures within the reservoir zone. This will enhance the effectiveness of the reservoir stimulation and improve the production capacity.

## 5. Conclusions

In this study, a three-dimensional hydraulic fracturing numerical simulation based on the cohesive element was established using the ABAQUS 2022 software to investigate the pore pressure distribution, fracture initiation and propagation, and fracture morphology during the fracturing process and to analyze the reasons why the fracture breaks through the upper and lower layers.

The pore pressure exhibited a stepped distribution around the fracture and an elliptical distribution in the vicinity of the fracture tip. The pore pressure in the reservoir was higher than that in the barrier, and the length of the fracture was much greater than its height and width.

The pore pressure at the injection point initially increased and then decreased before stabilizing over time. The fracture width at the injection point gradually increased in a curve pattern, fluctuating slightly after the fracture initiation and reaching the maximum width before gradually decreasing and stabilizing.

The fracture broke through the upper and lower layers at 10 s because the interlayer principal stress difference dominated the fracture expansion mode. The minimum horizontal principal stress difference was 3.5 MPa, which was less than 4 MPa, leading to the hydraulic fractures penetrating through the lithological interface and breaking through the upper and lower layers, which was not conducive to fracture height control and improvement of the fracturing effect.

The research findings have far-reaching implications for oil and gas exploration, as well as deep engineering applications such as deep mining, tunnelling, and HDR geothermal energy. We highlight its significance in advancing green energy and promoting sustainable practices. The insights gained from our study can serve as a valuable foundation for future research and practical applications in similar shale gas reservoirs and other deep engineering projects. For instance, the numerical modeling techniques and fracture propagation understanding developed in this study can be applied to different geological formations and reservoir conditions, allowing for a more accurate assessment of hydraulic fracturing operations in diverse contexts. Additionally, the knowledge and methodologies derived from this work can be utilized to optimize fracture design and improve production efficiency in various deep engineering applications, such as deep mining, tunnelling, and HDR geothermal energy. By discussing the potential extensions and applications of this work, we provide a roadmap for further research and practical implementations. This not only enhances the scientific value of our study but also strengthens its practical relevance and impact on the broader sustainability goals of the oil and gas industry and other deep engineering sectors.

**Author Contributions:** Conceptualization, X.Y. and Y.R.; methodology, Z.S.; software, F.C.; validation, S.C.; formal analysis, T.Z.; investigation, Y.C.; resources, J.S.; data curation, S.S.; writing—original draft preparation, S.S.; writing—review and editing, S.S.; visualization, S.S.; supervision, S.S.; project administration, S.S.; funding acquisition, S.S. All authors have read and agreed to the published version of the manuscript.

**Funding:** This study was supported by the Major Special Project of the Ministry of Science and Technology of PetroChina (Grant no. 2022DJ8004).

**Institutional Review Board Statement:** Not applicable.

**Informed Consent Statement:** Not applicable.

**Data Availability Statement:** Data are contained within the article.

**Conflicts of Interest:** Authors Shasha Sun, Zhensheng Shi, Feng Cheng, Tianqi Zhou, Yan Chang, and Jian Sun were employed by the PetroChina Research Institute of Petroleum Exploration and Development. Author Yun Rui was employed by the PetroChina Zhejiang Oil and Gas Field Company. The remaining authors declare that the research was conducted in the absence of any commercial or financial relationships that could be construed as a potential conflict of interest. The authors declare that this study received funding from Major Special Project of the Ministry of Science and Technology of PetroChina (2022DJ8004). The funder was not involved in the study design, collection, analysis, interpretation of data, the writing of this article, or the decision to submit it for publication.

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
