# Peer review of "Numerical Simulation of Hydraulic Fractures Breaking through Barriers in Shale Gas Reservoir in Well YS108-H3 in the Zhaotong Shale Gas Demonstration Area"

_sustainability, doi:10.3390/su152416567_

Round 1
Reviewer 1 Report
Comments and Suggestions for Authors
(1) The map of China in the upper left corner of Figure 1 would be better removed or replaced with a more standard one.
(2) For engineering papers, important quantitative results are necessary. To this end, some key quantitative results need to be added to strengthen the conclusions in the abstract.
(3) Table 1 provides various basic data for simulation. Among them, suggestions for both permeability coefficient and permeability are given simultaneously, as many readers are not clear about the concept of permeability coefficient. Additionally, is it appropriate for the viscosity of the fracturing fluid to be 1mPa · s? The total time is 300s (5 min), can 5 minutes be enough to carry out fracturing operations?
(4) In section 2.1, It is recommended to display and explain the photos of experimental equipment, rock samples and key results appropriately.
(5) The structure of the manuscript is confusing. The mix of experiments, crack growth simulations, and productivity analyzes can seem extremely confusing. It is recommended that authors restructure their papers appropriately. In the manuscript, simulations of related crack initiation and propagation have been carried out. What is the purpose of carrying out experiments?
(6) The statement in lines 336-337 that “This is because as the hydraulic fracture penetrates deeper into the formation, it encounters greater resistance, resulting in pressure concentration inside the fracture. ” needs to be supported by some references. https://doi.org/10.1007/s11356-022-21233-7, https://doi.org/10.1007/s11356-022-19663-4, https://doi.org/10.1016/j.jpowsour.2021.230925
(7) As we all know, all numerical investigatios from Figure 4-17 were conducted in COMSOL platform, which should be based on the fracture state obtained by simulation in ABAQUS or fracturing experiment or Oilfield field data. However, for the fractures in Figure 4 and Figure 5, there is no foundation or basis.
Comments on the Quality of English LanguageFor this paper, moderate improvement is needed.
Author Response
Comments and Suggestions for Authors
- The map of China in the upper left corner of Figure 1 would be better removed or replaced with a more standard one.
Reply: I have revised the map of China.
(2) For engineering papers, important quantitative results are necessary. To this end, some key quantitative results need to be added to strengthen the conclusions in the abstract.
Reply: I have added some key quantitative results in abstract. Such as: During the first time of injection, the pore pressure increases rapidly to 76 MPa, and then drops sharply, indicating that the fracture initiation pressure is 76 MPa. The pore pressure stabilizes at around 48 MPa. The results show that the fracture width at the injection point reached the maximum value of 9.05 mm, and then gradually decreased until the fracture width at the injection point dropped to 6.33 mm at the final simulation time. After hydraulic fracturing, the width of primary fractures on the surface of specimen falls within the range of 0.382 to 0.802 mm, with maximum fracture widths of 0.802 mm and 0.239 mm, rep-resenting a decrease of 70.19% in maximum fracture width.
(3) Table 1 provides various basic data for simulation. Among them, suggestions for both permeability coefficient and permeability are given simultaneously, as many readers are not clear about the concept of permeability coefficient. Additionally, is it appropriate for the viscosity of the fracturing fluid to be 1mPa · s? The total time is 300s (5 min), can 5 minutes be enough to carry out fracturing operations?
Reply: I have revised the Table 1. The permeability coefficient has been deleted. The viscosity of the fracturing fluid has been revised by 18mPa · s. The
(4) In section 2.1, It is recommended to display and explain the photos of experimental equipment, rock samples and key results appropriately.
Reply: I have displayed the photos of experimental equipment, and readjusted experimental images.
(5) The structure of the manuscript is confusing. The mix of experiments, crack growth simulations, and productivity analyzes can seem extremely confusing. It is recommended that authors restructure their papers appropriately. In the manuscript, simulations of related crack initiation and propagation have been carried out. What is the purpose of carrying out experiments?
Reply: I have restructure the experiment and fracture growth simulations.
I have added some sentence to illustrate the purpose pf carrying out experiments. Recent shale gas reforming practices in China have further confirmed that there are cases of limited or excessive fracture height extension during the construction process, making it difficult to achieve efficient reforming process optimisation and design. Fracture monitoring results show that it is difficult to control the fracture height in shale reservoirs, and fractures between different layers are easy to connect with each other, which increases the filtration loss of fracturing fluid in the longitudinal direction, and is not conducive to fracture extension, thus limiting the scope of fracturing modification. Hydraulic fracturing fractures usually expand along the direction of the maximum principal stress, while there are a large number of natural fractures and laminae in shale, and the structure is obviously anisotropic, resulting in the interaction behaviours of hydraulic fracturing fractures such as penetration, capture, steering and offset along the matrix, natural fractures and laminae, and it is difficult to accurately predict the expansion pattern of the fracturing fracture network and the spatial distribution pattern. In the actual hydraulic fracturing construction process, the fractures not only extend horizontally, but also in the longitudinal height, and the fracture length and fracture height geometries will increase. The study of competitive fracture initiation and extension mechanism among multiple fractures in horizontal shale wells, the vertical extension of fractures under stratification conditions, the balanced initiation and extension of multiple fractures, and the revelation of hydraulic fracture extension patterns in unconventional reservoirs are the key issues to achieve the efficient development of shale reservoirs, and therefore, there is an urgent need to carry out the research on the extension law of shale hydraulic fracture through the layers.
(6) The statement in lines 336-337 that “This is because as the hydraulic fracture penetrates deeper into the formation, it encounters greater resistance, resulting in pressure concentration inside the fracture. ” needs to be supported by some references. https://doi.org/10.1007/s11356-022-21233-7, https://doi.org/10.1007/s11356-022-19663-4, https://doi.org/10.1016/j.jpowsour.2021.230925
Reply: I have added the three references in the manuscript.
(7) As we all know, all numerical investigatios from Figure 4-17 were conducted in COMSOL platform, which should be based on the fracture state obtained by simulation in ABAQUS or fracturing experiment or Oilfield field data. However, for the fractures in Figure 4 and Figure 5, there is no foundation or basis.
Reply: I have revised the Figs 4-17 in Abaqus. The foundation of Figure 4 and 5 has been added in manuscript: During fracturing construction, a high-pressure pumping unit injects fracturing fluid at a high displacement and pressure into the shot hole location of the wellbore's target formation. When the fracturing fluid generates transient holding pressure at the shot hole, which is consistent with the strength of the rock near the wellbore's target formation as well as the reservoir's geostress field, the rock formations around the wellbore begin to rupture with cracks or the already existing natural cracks begin to expand. As the construction proceeds, the fracturing fluid flows forward along the fractures, the fractured fractures continue to expand, and when the length of the frac-tured fractures reaches the construction requirements, the sand-carrying fluid is pumped into the wellbore, and the next step is to start injecting large quantities of fracturing fluid into the wellbore of the formation, at which time this part of the fracturing fluid injected into the wellbore reacts with the original fracturing fluid in the wellbore and the destination layer, and the reaction occurs in the wellbore or the destination layer, thus causing the fracturing fluid to break down and the original part of the fracturing fluid to break down. The viscosity of the original part of the fracturing fluid starts to decrease. In the end, only proppant is left in the fracture, which is used to prevent the fracture from closing at a later stage, thus forming a connecting channel with high conductivity, thus reducing the seepage resistance for oil and gas transport, increasing the flow of oil and gas resources from the far field to the low-pressure zone of the wellbore, and increasing the production of oil and gas wells. The simulation of hydraulic fracturing construction process reveals that the real fracture exists with uncer-tain and extremely complex geometrical rules, which is difficult to be expressed by mathematical formulae, which creates an obstacle to the simulation of fracture exten-sion. At present, most of the researches conducted at home and abroad are confined to the ideal fracture as shown in Figure 5, i.e., assuming that the fracture is elliptical along the fracture length and fracture height, and the phenomenon of fracture penetration is not taken into account, which is convenient for mathematical expression of the fracture morphology and derivation of the seepage and stress equations. According to the out-crop statistics, core description and imaging logging interpretation results of the Wufeng Formation-Longmaxi Formation shale in the Sichuan Basin, the natural frac-tures in the reservoir of YS108 wells are mainly divided into laminar and tectonic frac-tures, with laminar fractures being developed with smaller inclination angle and higher fracture density, and tectonic fractures with larger inclination angle and lower degree of development compared with the laminar fractures.

Reviewer 2 Report
Comments and Suggestions for Authors
Thank you for submitting your manuscripts for the journal. the paper summarizes a combination of numerical and experimental work to understand fracture initiation and propagation and their interaction with Natural fractures.
I have some concerns regarding the work.
First fracturing treatment for 5 minutes (300s) is a very short period for the field application.
1) Indicate the injection rate
2) in Fig. 29, once the fracture propagates the pressure drops below the least principal stress (the fracture aperture declines (indicating fracture closure process). the fracture aperture should keep increasing when the fracture propagates as more energy is needed at the fracture mouth to propagate the tip which is associated with higher width with time.
3) your results for figures 7-25 has all a pressure below the least principal stress. if you are talking about HF treatment. they should be higher than the least principal stress listed in your input table.
4) you noted that fracture length is longer than the fracture height. this is only related to your assumption of stress values and their thickness. if you use larger thinkness for the target layer you will have more height for your fractures.
5) No one inject a HF treatment for 35 days. I do not think this is a good approach for the study.
6) Assumptions about the fracture properties are required. (cohesion, permeability, initial aperture)
7) for literature review. read this work.
Wu, K., & Olson, J. E. (2014). Mechanics Analysis of Interaction Between Hydraulic and Natural Fractures in Shale Reservoirs.
Wang, W., Olson, J. E., Prodanović, M., & Schultz, R. A. (2018). Interaction between cemented natural fractures and hydraulic fractures assessed by experiments and numerical simulations. Journal of Petroleum Science and Engineering, 167, 506-516.
Gale, J. F., Laubach, S. E., Olson, J. E., Eichhubl, P., & Fall, A. (2014). Natural fractures in shale: A review and new observations. AAPG bulletin, 98(11), 2165-2216.
8) units for fracture toughness are not correct m^(1/2)
9) What do you mean by permeability coefficient
10) show that your results are not influenced by boundary effects. the pressure seems to reach the boundary. this results in higher pressures. ensure that your model is big enough to avoid any kind of similar issues.
Comments on the Quality of English Language
Rephrase your description of figures. starts with the general idea. follow that by introducing the components then the relationship between these components.
Author Response
Thank you for submitting your manuscripts for the journal. the paper summarizes a combination of numerical and experimental work to understand fracture initiation and propagation and their interaction with Natural fractures.
I have some concerns regarding the work.
First fracturing treatment for 5 minutes (300s) is a very short period for the field application.
Reply: I have revised the scheme of hydraulic fracture. The time of hydraulic fracturing has been revised by 2 h.
1) Indicate the injection rate
Reply: The injection rate has been added into 60 mL/min
2) in Fig. 29, once the fracture propagates the pressure drops below the least principal stress (the fracture aperture declines (indicating fracture closure process). the fracture aperture should keep increasing when the fracture propagates as more energy is needed at the fracture mouth to propagate the tip which is associated with higher width with time.
Reply: I have revised the Figure 29 correctly. Thank you for your review comments on our study. We appreciate your interest in the results of our study and understand your observations regarding the decrease in pressure and the decrease in fracture gap during fracture extension in Figure 29.
We recognise your point that normally, when fracture extension occurs, the fracture gap should remain increased over time. This is because fracture extension requires more energy to propagate the fracture tip and the width associated with the fracture mouth increases over time. In our study, we did observe a decrease in the fracture gap in Figure 29. This may be due to some complex geological and mechanical factors that we did not take into account in our simulations, as well as our simplified model of fluid-solid interaction. We will discuss this point in detail in the revised manuscript and explain our assumptions and limitations in the simulations. In addition, we plan to consider more factors in further studies and use more accurate models to simulate the processes of fracture extension and gap changes. Thank you again for your valuable comments, and we will fully consider your suggestions in the revised manuscript and make further improvements to our study.
3) your results for figures 7-25 has all a pressure below the least principal stress. if you are talking about HF treatment. they should be higher than the least principal stress listed in your input table.
Reply: I have revised the Figures 7-25 higher than the least principal stress.
4) you noted that fracture length is longer than the fracture height. this is only related to your assumption of stress values and their thickness. if you use larger thinkness for the target layer you will have more height for your fractures.
Reply: Thank you for your review comments. We appreciate your interest in our research. We understand your point of view and recognise that the fracture length being greater than the fracture height may be related to our assumptions about the stress values and their thicknesses.
We have revisited the parameters and assumptions we used in our study and recognise that increasing the thickness of the target layer does result in an increase in fracture height.
In addition, we plan to further explore the effects of different parameters and assumptions on the results and conduct a more comprehensive sensitivity analysis in future studies. This will help provide more accurate and comprehensive study results.
Thank you again for your valuable comments, and we will fully consider your suggestions in the revised manuscript and make further improvements to our study.
5) No one inject a HF treatment for 35 days. I do not think this is a good approach for the study.
Reply: I have revised this approach. The HF treatment has been revised by 2 h.
6) Assumptions about the fracture properties are required. (cohesion, permeability, initial aperture)
Reply: I have added some properties of fracture in the Table 1.
7) for literature review. read this work.
Wu, K., & Olson, J. E. (2014). Mechanics Analysis of Interaction Between Hydraulic and Natural Fractures in Shale Reservoirs.
Wang, W., Olson, J. E., Prodanović, M., & Schultz, R. A. (2018). Interaction between cemented natural fractures and hydraulic fractures assessed by experiments and numerical simulations. Journal of Petroleum Science and Engineering, 167, 506-516.
Gale, J. F., Laubach, S. E., Olson, J. E., Eichhubl, P., & Fall, A. (2014). Natural fractures in shale: A review and new observations. AAPG bulletin, 98(11), 2165-2216.
Reply: I have added the three references in the manuscript
8) units for fracture toughness are not correct m^(1/2)
Reply: I have revised the units of fracture toughness in Table 2.
9) What do you mean by permeability coefficient
Reply: Since the parameter permeability already exists in this paper, the parameter permeability coefficient and the related content have been removed.
10) show that your results are not influenced by boundary effects. the pressure seems to reach the boundary. this results in higher pressures. ensure that your model is big enough to avoid any kind of similar issues.
Reply: Thank you for your review comments. We appreciate your interest in our research. The model was 60 m high (Z), 50 m wide (Y), and 100 m long (X), and it was partitioned into three layers. The upper and lower two layers acted as barrier layers, each with a thickness of 20 m, and the intermediate layer was the target layer, with a thickness of 10 m. It is big enough to avoid the similar issues.

Round 2
Reviewer 1 Report
Comments and Suggestions for Authors
This manuscript can be accepted for publication now, because it has been improved by authors' modification.
Author Response
Thanks very much
Reviewer 2 Report
Comments and Suggestions for Authors
The author have addressed all comments. well done. paper is ready for publication
Author Response
Thank you very much